



# Ocean bubbles under high wind conditions. Part 2: Bubble size distributions and implications for models of bubble dynamics

Helen Czerski[1], Ian M. Brooks[2], Steve Gunn[3], Robin Pascal[4], Adrian Matei[1], Byron Blomquist[5,6]

[1]Department of Mechanical Engineering, University College London, London, WC1E 7BT, UK
[2]School of Earth and Environment, University of Leeds, Leeds, LS2 9JT, UK
[3]Department of Electronics and Computer Science, University of Southampton, SO17 1BJ, UK
[4]National Oceanography Centre, Southampton, SO14 3ZH, UK
[5]Cooperative Institute for Research in Environmental Sciences, University of Colorado, Boulder, CO, USA
[6]NOAA Physical Sciences Laboratory, Boulder, Colorado, USA

*Correspondence to*: Helen Czerski (h.czerski@ucl.ac.uk)

**Abstract.** Bubbles formed by breaking waves in the open ocean influence many surface processes but are poorly understood.
We report here on detailed bubble size distributions measured during the High Wind Speed Gas Exchange Study (HiWinGS) in the North Atlantic, during four separate storms with hourly averaged wind speeds from 10-27 m s$^{-1}$. The measurements focus on the deeper plumes formed by advection downwards (at 2 m depth and below), rather than the initial surface distributions. Our results suggest that bubbles reaching a depth of 2 m have already evolved to form a heterogeneous but statistically stable population in the top 1-2 metres of the ocean. These shallow bubble populations are carried downwards by
coherent near-surface circulations; bubble evolution at greater depths is consistent with control by local gas saturation, surfactant coatings and pressure. We find that at 2 m the maximum bubble radius observed has a very weak wind speed dependence and is too small to be explained by simple buoyancy arguments. For void fractions greater than 10$^{-6}$, bubble size distributions at 2 m can be fitted by a two-slope power law (with slopes of -0.3 for bubbles of radius < 80 μm and -4.4 for larger sizes). If normalised by void fraction, these distributions collapse to a very narrow range, implying that the bubble
population is relatively stable and the void fraction is determined by bubbles spreading out in space rather than changing their size over time. In regions with these relatively high void fractions we see no evidence for slow bubble dissolution. When void fractions are below 10$^{-6}$, the peak volume of the bubble size distribution is more variable, and can change systematically across a plume at lower wind speeds, tracking the void fraction. Relatively large bubbles (80 μm in radius) are observed to persist for several hours in some cases, following periods of very high wind. Our results suggest that local gas supersaturation around the
bubble plume may have a strong influence on bubble lifetime, but significantly, the deep plumes themselves cannot be responsible for this supersaturation. We propose that the supersaturation is predominately controlled by the dissolution of bubbles in the top metre of the ocean, and that this bulk water is then drawn downwards, surrounding the deep bubble plume





and influencing its lifetime. In this scenario, oxygen uptake is associated with deep bubble plumes, but is not driven directly by them. We suggest that as bubbles move to depths greater than 2 m, sudden collapse may be more significant as a bubble

destruction mechanism than slow dissolution, especially in regions of high void fraction. Finally, we present a proposal for the processes and timescales which form and control these deeper bubble plumes.

## 1. Introduction

The heterogeneous bubble plumes produced in the open ocean during high wind conditions have been studied for many years

(Medwin and Breitz, 1989; Farmer et al., 1993; Graham et al., 2004; Vagle et al., 2010). These plumes are thought to enhance air-sea gas transfer (Wanninkhof, 2014; Farmer et al., 1993; Woolf et al., 2007) and to change the optical (Stramski and Tegowski, 2001) and acoustical (Deane, 2016; Trevorrow, 2003; Van Vossen and Ainslie, 2011) properties of the near-surface ocean. The visible foam patches associated with wave breaking, known as whitecaps, eject aerosol particles into the atmosphere as the bubbles burst (De Leeuw et al., 2011). However, the challenges associated with following rapid, small-scale processes

in the top few metres of stormy seas mean that we still lack a complete description of bubble evolution and dynamics.

Much of the literature has focussed on the processes of wave breaking because this is the source of the bubbles, and because short-lived large bubbles associated with high void fractions are thought to be particularly important for $CO_2$ transfer from atmosphere to ocean (Farmer et al., 1993). Wave-breaking is often accompanied by the formation of deep bubble plumes (>

~2 m) which are easily observed using sonar. These are known to vary with environmental conditions (Vagle et al., 2010), and have been clearly associated with Langmuir circulation patterns (Zedel and Farmer, 1991). However, the likely path of an individual bubble, its size evolution, and the associated timescales are not yet clear. These deep plumes are thought to be important for the uptake of poorly soluble gases like oxygen, and recent work (Atamanchuk et al., 2020) suggests they might be critical for the export of oxygen to the deep ocean. Much of the literature on these plumes focuses on bubble presence and

plume description, and the challenging task of understanding the detailed processes occurring within the observed structures still remains. The ultimate goal is to clarify the mechanisms linking location, radius, and timescale as a bubble progresses from formation to destruction.

It has proven challenging to develop a robust relationship between sea state, water conditions and a quantitative description of subsurface bubble plumes. The lack of detailed data from the open ocean is a significant limitation, especially at wind speeds

above 15 m s$^{-1}$ and when swell is present. The practical difficulties of making measurements in the open ocean have led to a wide range of laboratory studies in wave tanks, usually in fresh water (Rojas and Loewen, 2010; Anguelova and Huq, 2012; Leifer and De Leeuw, 2006; Lamarre and Melville, 1991; Blenkinsopp and Chaplin, 2007), and less often in salt water (Blenkinsopp and Chaplin, 2011; Callaghan et al., 2016, 2017). It is known that the presence of salt changes bubble size distributions by preventing bubble coalescence (Kolaini, 1997; Slauenwhite and Johnson, 1999). Although useful, the results

of laboratory experiments are hard to generalise because the physical processes involved (bubble fragmentation, turbulence and wave breaking parameters) are not easily scalable (Deane et al., 2016), and natural wave breaking is a three-dimensional



process, while laboratory tank studies typically constrain the system to two dimensions. Some modelling studies have also been done (Fraga and Stoesser, 2016), but current numerical models cannot yet reproduce the complexity of this multi-phase flow with sufficient detail to draw strong conclusions. The combination of open ocean and laboratory experiments has produced a general overview of the generation and development of bubble plumes immediately following on from breaking waves, but details at the scale of individual bubbles are lacking.

Most open-ocean breaking waves are spilling rather than plunging (Deane and Stokes, 2002). As the breaking wave crest overturns, air is trapped in a region of highly turbulent water and a distinctive initial bubble size distribution is created within the first second after breaking. Void fractions in the actively breaking crest exceed 0.1 (Lim et al., 2015; Deane and Stokes, 2002). A critical threshold in this process, known as the Hinze scale, denotes the bubble size at which the restoring force caused by surface tension balances the distorting turbulent shear forces and therefore the smallest bubble size that the turbulence can fragment. The Hinze scale is thought to vary only between 0.7 and 1.7 mm over two orders of magnitude of wave energy, because the maximum turbulent dissipation rate appears to saturate beneath breaking waves (Deane et al., 2016). Above this size turbulence causes bubble fragmentation, and the bubble size distribution has a power law dependence on radius with a slope of -10/3 (Garrett et al., 2000; Deike et al., 2016). Deike et al (Deike et al., 2016) used a combination of laboratory experiments and theoretical assumptions to generate a model for the bubble size distribution under the active crest of a breaking wave, which applies to bubbles above the Hinze scale and covers the majority of the void fraction during active breaking. Bubbles smaller than the Hinze scale are formed by Messler entrainment, and jet and drop impact (Lim et al., 2015), although these processes are not well-understood. The slope of the bubble size distribution below the Hinze scale is observed to be approximately -1.5, but the smallest radius to which the slope extends is unclear. There are still many open questions associated with this initial period of bubble formation, particularly the variability of the smaller bubbles, and the dependence of the bubble formation processes on temperature and surfactant load.

Once formed, bubbles move due to buoyancy and advection. Anguelova and Huq (Anguelova and Huq, 2012) observed very early bubble plumes moving forwards at half the dominant wave phase speed. Small bubbles may be advected by Langmuir circulation, acting as tracers for convergence zones (Thorpe, 1982; Thorpe et al., 2003; Zedel and Farmer, 1991), and may also act to suppress turbulence in those regions (Gemmrich, 2012). Vagle et al. (Vagle et al., 2012) show that a high heat flux appears to influence near-surface bubble distribution, with near-surface turbulence reduced by a factor of 10 during periods with high downward heat flux. They also found some evidence that numbers of large bubbles (>200 μm in radius) at a depth of 0.5 metres might be different during periods of positive and negative surface heat flux.

## 1.1 Bubble size distributions

Once the initial bubble size distribution is established, it will steepen at the large end as bubbles rise to the surface (Garrett et al., 2000) and is expected to flatten at the small end, because small bubbles are likely to dissolve faster than larger ones (depending on their coating of surfactants and particulates), although there is no direct evidence for this in the ocean. The bubbles in the middle of this range may be used as tracers for water movement. Open ocean bubble size distributions at various





depths have been collected by de Leeuw and Cohen (Leeuw and Cohen, 2002)(photographic, 1-3 m), Terrill et al. (Terrill et al., 2001)(acoustical methods, 0.7 m), Deane and Stokes (Deane and Stokes, 2002)(photographic, 0.3 m), Vagle et al. (Vagle et al., 2010; Vagle et al., 2012) (acoustical resonators, 0-5.5 m), Norris et al. (Norris et al., 2012) (photographic, 0.4 m) and Randolph et al. (Randolph et al., 2014) (optical scattering, 6-9m). The Randolph study is notable for a bubble size measurement

range from 0.5-125 μm radius, although the deployment site was only a few metres from the ship. This study did not observe a peak in the bubble size distribution, noting significant bubble numbers with radii < 10 μm.

Deane et al. (Deane et al., 2013) constructed a partial model for the larger bubbles forming a persistent surface bubble layer (radii > 100 μm), based on the idea that bubbles will be trapped in the surface layer if their buoyant rise speed does not exceed the turbulent flow speed expected at a given wind speed. This model was designed for the evaluation of the acoustics of the

bubbly water near the surface and did not contain an explicit bubble source function, but matched observations of acoustical attenuation at sea.

Crawford and Farmer (Crawford and Farmer, 1987) noted that there is a persistent layer of bubbles near the surface at high winds, down to approximately 10 m. They hypothesised that although the deep bubble plumes vary in time and space, there may be an equilibrium average bubble distribution for a given set of conditions, where the bubble sources and sinks balance.

We are only aware of one detailed empirical model for bubble size distribution inside the deeper plumes, constructed by Vagle et al. (Vagle et al., 2010) using acoustical resonators at different depths in wind speeds from 12-23 m s$^{-1}$ and averaged bubble size distributions. Most studies have focussed on quantifying the features of individual deep bubble plumes - depth, persistence and number - rather than the averaged bubble field.

In summary, there is very little *in situ* evidence on the processes advecting and altering bubbles after the active part of the

breaking wave. To make progress on the open questions about the importance of deep plumes, particularly for oxygen uptake, a clear understanding of the dominant processes and timescales is essential.

Here we present bubble size distributions measured during the High Wind Speed Gas Exchange Study (HiWinGS), in the North Atlantic Ocean in 2013. Measurements were made using a custom-built bubble camera, acoustical resonators, and an upward-looking sonar mounted on an autonomous spar buoy during four storms, with a range of hourly-averaged wind speeds

from 10-27 m s$^{-1}$. We address specific questions about the mechanisms driving bubble presence and influence: how and when bubbles are transported downwards from the surface, how the size and number of bubbles varies with conditions, the overall path of a bubble through the water column, and the mechanism and manner of its destruction. We have used the term "shallow populations" for the near-surface bubbly regions formed by every breaking wave, and "deep plume" for the water parcels with void fractions of 10$^{-6}$ or more which are advected downward by coherent flow structures to 2 m depth and below. A companion

paper (Czerski et al., 2021) describes the larger scale plume structures studied using void fraction as a metric. At the end of this paper we use the results from both papers to present a suggested outline of the bubble processes leading to deep bubble plumes.

**Methods**





The HiWinGS cruise took place between 9 October and 14 November, 2013, on board the R/V *Knorr*. Blomquist et al.
(Blomquist et al., 2017) provide an overview of the entire cruise and the main gas transfer results. Here we focus on
measurements made from an 11-m free-floating spar buoy (Pascal et al., 2011). The buoy carried an upward-pointing sonar,
acoustical resonators at 6 m and 4 m depth, an acoustic Doppler velocimeter (ADV), a specialised bubble camera at 2 m depth,
capacitance wave wires, and a downward-pointing foam camera mounted on the top of the platform. Full details of the

instruments and the conditions are provided in (Czerski et al., 2021). We follow the (Blomquist et al., 2017) station numbering
for our four deployments: 17–21 October (station 3), 24–26 October  (station 4), 1–3 November (station 6), and 7–9 November
(station 7).

The buoy was designed to orient into the wind and all bubble sensors were positioned on the upwind side. However, the data

on the relative water flow around the buoy showed that the buoy was being pushed downwind faster than the wind-induced
surface currents at the depth of the bubble sensors; this is discussed in detail in Czerski et al. (Czerski et al., 2021). We are
confident that the measurements taken are still representative of the water at their depth, but the buoy was moving through
bubble plumes in the downwind direction with speeds of 2-15 cm s$^{-1}$ rather than remaining stationary with respect to the water
at its base.


The bubble data at 2 m was collected by a custom-built bubble camera (Al-Lashi et al., 2018; Al-Lashi et al., 2016), taking
images at 15 Hz which were averaged to provide one bubble size distribution every second. The bubble radius measurement
range was from 20 μm to a few millimetres, and the camera operated continuously for blocks of 45 minutes at intervals of 3-
4 hours. The movement of the buoy due to the waves caused the instrument depths to vary with respect to the instantaneous

surface. At the highest wind speeds (above 20 m s$^{-1}$), the bubble camera was within 1 m of the surface approximately 10% of
the time, and within 0.5 metres of the surface approximately 2.5% of the time.

Acoustical resonators are a proven way of making bubble size distribution measurements down to void fractions of 10$^{-8}$.
(Medwin and Breitz, 1989; Czerski et al., 2011b; Czerski, 2012). Here they provided one size distribution every second,

covering a radius range of 5-200 μm. The acoustical resonator at 6 m did not provide usable data, but the resonator at 4 m
provided good data for every deployment except Station 4.

The buoy was deployed while the winds were rising at the start of each storm, and it then floated freely for 3-5 days until the
storm had passed and recovery was possible. We show data from four deployments with wind speed ranges of 6-15 m s$^{-1}$, 8-

27 m s$^{-1}$, 10-19 m s$^{-1}$ and 9-18 m s$^{-1}$ respectively. A Datawell DWR-4G Waverider buoy was deployed during the same periods,
providing 2D wave spectra. Meteorological measurements were made from the foremast of the ship. Over the entire expedition,
we collected 29 hours of camera data and 52 hours of resonator data. The resulting bubble size distributions are the most
comprehensive data set yet collected on the bubbles found within the top few metres of the open ocean.



## 2. Results

Measured void fractions at a depth of 2 m ranged from $10^{-9}$ to $10^{-4.5}$, with a sharp cut off at the higher limit; detailed descriptions of void fraction results are given in Czerski et al. (Czerski et al., 2021). Void fractions at 4 m varied from $10^{-8}$ (the noise level) to $2 \times 10^{-7}$, rising above the noise for approximately 10% of the overall measurement time. We did observe "plumes" (we use the term here to indicate bubbly regions several metres in size with void fractions at 2 m which were above $10^{-6}$), but there

was a heterogeneous background layer of bubbles present at 2-m depth in all conditions. The void fraction probability distributions were smooth and varied with conditions, and there were no other criteria that could separate a "plume" from the background bubble field at 2 m.

### 2.1 Maximum Bubble Radii

Figure 1 (a,b) shows the probability distributions of the maximum bubble radius at 2 m observed in each one second period, split by wind speed and void fraction. The maximum bubble size is tightly correlated with void fraction and has a more limited relationship with wind speed. Bubbles with a radius larger than 220 μm were rare at the camera depth, present in only 5% of the images even at the highest wind speeds, and only ever during the periods when the void fraction was above $10^{-6.5}$.  Figure 1 (c) shows the radius at the 90th, 95th, 99th and 100th percentiles of the probability distribution of the maximum bubble radius

across the entire data set, segregated by wind speed. For 99% of the images at all wind speeds, the maximum bubble radius was 300 μm or below.  The largest bubble observed at any point is 500 μm in radius and at the lowest wind speeds; it seems likely that these very large bubbles were not observed at the highest wind speeds only because there is less data available in those conditions. Discounting the top 1% (which could be due to the camera being temporarily very close to the surface or a large co-located breaking waves), it is striking that there is very little wind speed dependence in the maximum bubble radii.


The possible constraints on the maximum bubble size at a given depth are bubble production mechanism and rate, buoyancy, flow structures (for example, turbulence, convection, or Langmuir circulation) and dissolution or destruction processes (which depend on the water saturation state and the bubble coating). Deane et al. (Deane et al., 2013) used a limited model to estimate the maximum expected bubble size based on the assumption that bubbles will persist in the near-surface layer when the rms

vertical velocity fluctuations due to turbulence are comparable to or greater than the bubble rise speed due to buoyancy. Those predictions are shown in Fig. 1 (c) for 2 m depth, and suggest that the theoretical maximum bubble radius varies from 50 μm (at $U_{10} = 3$ m s$^{-1}$) to 700 μm (at $U_{10} = 20$ m s$^{-1}$).  Our results do not follow the predicted pattern, although the probability distribution of maximum bubble size does show some variation with wind speed, as shown in Fig. 1 (a). This opens up the possibility that the major constraint on maximum bubble size at a given depth may not be buoyancy (discussed further in Sect.

3.2). However, the observed pattern could also be due to effects which are only apparent when the full complexity of near-surface turbulence is included in the model (the relative simplicity of the model is acknowledged in Deane's paper).




## 2.2 Bubble size distributions

Before considering the bubble size distributions, we note that an artefact arises when time averaging 1 Hz bubble size distribution measurements over long periods. The artefact is an artificial steepening of the averaged bubble size distributions at the high radius end, and it is discussed in detail in Appendix A. What this feature obscures is that the instantaneous bubble size distributions to the right of the slope break are straight lines with no steepening. Consequently, the instantaneous distributions should be used for understanding bubble dynamics, not the averaged distributions. For this reason we focus on
the 1 Hz measurements here, without time averaging.

Figure 2 (a) shows all the bubble size distributions measured in all conditions for both camera and resonator, with an individual bubble size distribution plotted for every second. At any given radius, R, this concentration varies by a factor of 25-30 at 2 m depth, and a factor of 10-20 at 4 m. Figure 2 (b) shows the same data, but each individual bubble size distribution has been
normalised by its void fraction. This collapses the data, reducing the range by approximately a factor of 5 at 2-m and a factor of 8 at 4-m. The normalised size distributions at 2 m have a broadly consistent shape, which can be fitted as two straight lines with a slope break at approximately R = 80 μm. Below the break the slope is -0.4 to -0.6, while above it the slope is much steeper, at -3.8 to -5.0. The void fraction normalisation collapses the bubble size distributions to a much narrower range in all cases except those with very low bubble numbers. This implies that the bubble size distribution is relatively stable, and that
variations in void fraction are dominated by this stable population diffusing outward in space rather than individual bubbles changing size.

Splitting the 2-m size distributions by void fraction reveals a more systematic pattern (Fig. 3). The normalised bubble size distribution is highly dependent on void fraction: the spread is large at low void fractions, and they cluster tightly at void
fractions above $10^{-6}$. The black lines (identical on all subplots) have slopes of -4.4 and -0.3, and there is a factor of 4 between the two lines (a halving and doubling from a central line, representative of the mean distribution and not shown). A quantitative assessment of how universal the fit is can be made by considering how many of the points on each individual bubble size distribution fit between the black lines. For bubble size distributions with a void fraction between $10^{-5}$ and $10^{-4.5}$, 61% of the one-second distributions have 85% of their points between these bounds, showing very high uniformity. The statistics for all
void fraction ranges are shown in Appendix B.
An alternative way of viewing this data is shown in Fig. 4 (a), where the mean bubble size distributions for each void fraction are normalised by void fraction, again across all deployments and conditions. The distributions are again tightly clustered for void fractions above about $10^{-6.5}$. Figure 4 (b) shows the volume distribution for each of the average size distributions; it is striking that for all void fractions between $10^{-7}$ and $10^{-4.5}$, the peak volume occurs close to a bubble radius of 80 μm. The radius
at the peak volume will be examined in more detail in Sect. 2.3.





The bubble size distribution data from 4 m (Fig/ 2) show a steep slope of -3.1 to -3.6 and lack an unambiguous slope break. The acoustical data is harder to interpret for the smallest bubbles, because coatings will affect the acoustics (Czerski et al., 2011a), because the void fractions at 4 m are significantly lower than at 2 m, and because there is more noise in the data for

small radii.  There could be a slope break at a radius of 50 μm or less, but there is insufficient data to confirm this. Normalisation by void fraction also collapses the spread of the resonator data to a very narrow band (from a factor of 16 to a factor of 2). The overall void fractions at 4 m are less than those at 2 m by factors of up to 100, but the normalised bubble size distributions are very similar at the two depths. The range of observed void fractions is far narrower at 4 m, and the measurements rose above the noise level relatively rarely, so any patterns observed at that depth rest on weaker evidence.

These results show that although the measured void fraction at 2 m varied by a factor of $10^4$, the shape of the bubble size distribution associated with a particular void fraction is tightly constrained. We never observe larger bubbles (100-200 μm) without also seeing smaller bubbles present, even over a short (1 s) interval. This implies that the bubble sizes are well-mixed, and that there is no significant sorting process acting to separate bubbles of different sizes within our observed range. Bubbles are consistently present at 2 m right down to the smallest radius measured by the camera (20 μm), which implies that there is

no rapid dissolution process once they shrink below a critical size.

The implication is that for void fractions above $10^{-6.5}$ the size distribution isn't evolving (bubbles aren't growing or shrinking), but that the differences in void fraction are mainly due to bubbles being advected around the bulk water, gradually becoming more spaced out without changing their size, or are being destroyed by a mechanism that is independent of radius.


At void fractions below $10^{-6.5}$ the bubble size distributions do not collapse to a narrow band when normalised by void fraction. It appears that outside the higher void fraction regions, different mechanisms dominate the bubble size distribution which allow for more variation. These bubbles could be older (because they have been drifting in the surface water for longer), and may therefore have been exposed to a wider range of conditions for a longer time period, producing a variety of outcomes.

This raises the question of bubble longevity and how bubbles finally vanish. One critical question is whether bubbles change size once they have been submerged for more than a few minutes, and when and how that happens. A more detailed analysis of how the gas volume is distributed across bubbles of different radii can address that question, because a fitted peak volume is a more sensitive measure of small changes in bubble size.

**2.3 Volume peak fitting**

Gaussian fits were calculated for individual 1 s volume distributions for both camera and resonator data, in order to identify the radius at the volume peak. The fitting process provides better radius resolution than relying on the bin size responsible for the largest volume fraction. Full details of the fitting are given in Appendix C.


Figure 5 shows the radii of the volume peak positions at both 2 m and 4 m for one 45-minute period during wind speeds of 18 m s$^{-1}$. The largest radii at the peak volume are generally between 60 and 80 μm at both depths during this period. Peaks in void fraction generally coincide with a volume peak at a larger radius, but this does not exceed 80 μm for 10 second average values during this period. This is consistent with the normalised bubble size distributions discussed above, where bubble populations with a void fraction higher than $10^{-6.5}$ collapse to a very similar normalised distribution.

At 2 m, the peak volume has a limited relationship with the void fraction, and does not show a large decrease immediately after a peak.  If bubbles were shrinking with time as they dissolved, the radius of peak volume would consistently decline after a peak in void fraction, but here there is only limited evidence for this in Fig. 5 (a). It is not clear how well mixed these high void fraction regions are (see Sect. 3.2), and the continual buoy drift prevents straightforward separation of temporal and spatial changes. However, if bubbles were shrinking, it is unlikely that they would be immediately replaced by larger bubbles

in all cases, and so the consistent peak void fraction suggests that shrinking is limited.  Throughout all the higher wind speed periods with void fractions above $10^{-6.5}$, the radius of the volume peak generally remains very similar as void fraction rises and falls, although there is greater variability when void fractions are low.

There is a far more pronounced relationship at 4 m depth (Fig. 5 (c), (d)). The radius of the volume peak closely tracks the

void fraction, with the maximum 80 μm radius being reached only for the highest void fractions and the minimum possible fitted radius reached as the void fraction drops to the noise level.  The bubble numbers at 4 m only rose above the noise level for a small fraction of the time, but when they did, it was clear that each passing peak in void fraction was associated with an increase and then decrease in the volume peak radius.  The largest observed bubble radius at the volume peak is very similar at both depths.  However, the void fractions at 4 m are a factor of ~100 lower than those at 2 m, and also occupy a much

smaller spatial region. This suggests that the speed or mechanism of bubble destruction varies with depth and may have a weaker dependence on radius. Overall, the bubbles are smaller at 4 m depth than 2 m, but this is largely because they are smaller at the plume edges.

Figure 6 shows data from 2 m similar to Fig. 5 (a) and (b) but for a wider range of conditions. During higher winds and periods of high void fraction, the volume peak radii varied very little (Fig. 6 (a-b)). But during lower winds and periods of low void fraction (Fig. 6c-f; and also at 4 m, see Fig. 8), the volume peak radius increased significantly and then decreased as a plume advected past the camera. In Fig. 6 (e) (at low wind speeds of 10-12 m/s), the largest radius at the volume peak was the same as for far higher winds, but it increased and decreased as the plume advected past. It is also noticeable that the plume in this

example was relatively narrow: approximately 6 m wide given the buoy drift speed while other plumes seen in Fig. 6 were typically 30 metres wide. This change at low wind speeds suggests that gas saturation state may have a role to play, if plumes sit within locally saturated waters. The bubbles on the edge of a plume may shrink as they lose gas to their surroundings, while the region in the centre of a plume is more saturated and bubbles maintain their size for longer.



If bubble dissolution was a major influence on bubble size, the expected pattern would be a very quick rise in bubble peak radius as a plume was formed and then a slower decrease in the bubble size at peak volume. We do not see this pattern except at low void fractions, and in all cases the speed of rise and fall are very similar, suggesting that the observed patterns are due to spatial variation and not a bubble population which is changing over time.


Scatter plots of 10 s averages of volume peak radius against void fraction for each deployment at a depth of 2 m are shown in Fig. 7. There are clear differences between the deployments, which seem likely to be due to differing environmental conditions: surfactant load, temperature, the gas saturation state of the water, and possibly bubble production mechanisms. The data for station 6 (Fig. 7 (c)) shows a very clear upper limit to the volume peak radius, following two straight lines with a slope break

at a void fraction of ~$10^{-7}$. The same lines are shown on all other panels for reference. The straight lines imply that over each segment the maximum volume peak radius is proportional to the logarithm of the void fraction, with a slope break at $10^{-7}$. The fitted volume peak radii vary between 20 μm and 90 μm over the whole data set. In the two deployments with the lowest winds (station 3 (Fig 7a) and station 7 (Fig 7d)) the peak radii are generally lower than in the cases with higher winds. In a minority of cases, the fitting may produce a peak at 20 μm (the smallest size measured) when the real peak occurred at a lower

radius. However, this affects only a minority of cases. In general, when void fractions are higher than approximately $10^{-6.5}$, the peak volume position does vary with void fraction, but only over a small range (50-80 μm for a void fraction range of $10^{-6}$ to $10^{-4}$). Once the void fraction drops below $10^{-6}$, a far wider range of volume peak positions is seen in most cases. This is consistent with the normalised bubble size distributions discussed above: there is one basic shape for the bubble size distribution at void fractions higher than $10^{-6}$, but far greater variation below that level.


There are clear differences between the bubble population characteristics for each deployment. Segregation of the data by Hs, U10, the wind-wave Reynolds number ($Re_{ww}$) and oxygen saturation (shown in Appendix C) does not reveal convincing relationships between the population characteristics and those parameters. However, we note that our oxygen saturation measurements have poor time resolution and were not co-located with the buoy, and that a more thorough investigation of the

effect of gas saturation would require high time resolution gas saturation measurements that were co-located with the bubble sensors.

Figure 7 (a) shows the deployment with the lowest wind conditions: 6–15 m s$^{-1}$, without any storms in the days immediately preceding. Almost all the void fractions are lower than $10^{-6.5}$, and the volume peak is always between 20 and 40 μm. A

reasonable assumption is that the surface waters were not super-saturated before this storm (see Fig. 9), and that therefore dissolution processes are likely to have happened before a stable population was reached.



Figure 7 (b) shows data from the largest storm, with wind speeds between 7 and 27 m s$^{-1}$. No steep drop-off in volume peak is seen at the smaller void fractions, consistent with the idea that these are stable bubbles which aren't destroyed rapidly, but are

being advected through the surface water, spreading out in space but not changing significantly in size. There is a notable increase in peak volume bubble radius at very low void fractions; these data points are all from a specific time period. This occurred just after a very rapid drop in wind speed from 20 m s$^{-1}$ to 10 m s$^{-1}$ over the course of four hours as the eye of the storm approached (00:00 - 04:00 on 25 October). Although the void fractions were low during this period, the existence of large bubbles after four hours without breaking waves is clear evidence that a small number of large bubbles remained intact

without shrinking for several hours as the eye of the storm passed.

The third and fourth deployments follow a similar pattern to each other, with consistently smaller bubble sizes in the final deployment. This last deployment took place in far warmer waters in the Gulf Stream, and we cannot rule out the possibility that the temperature influenced the stable bubble size during that deployment.

Figure 8 shows the radius of the bubble volume peak for the resonator data at 4 m, for all deployments. One second data is shown here because the bubble events seen at 4 m are far shorter in general, and far fewer overall. At this depth, it is clear that individual patches of bubbles each follow their own pattern, since the progression for each individual plume can be seen. There is a very clear relationship between void fraction and the radius of peak volume for individual plumes, but that relationship varies between plumes. It is striking that the pattern appears to be limited by an envelope similar to the one seen

at 2 m in Fig. 7 (c).

### 2.4 Gas Saturation

We have very few direct measurements of gas saturation state. Figure 9 shows dissolved oxygen data from the CTD casts for the top ten metres of the ocean, and the extrapolated saturation state at the surface. The data in Fig. 9 (a) is shown as percentage

saturation for each specific depth (in contrast to the normal presentation of similar data, where oxygen saturation is expressed as a percentage of the surface saturation level). The distinction matters for understanding bubble dynamics because even an additional two metres of depth increases the saturation oxygen concentration significantly. The surface ocean was always undersaturated during our measurements, as expected for this time of year. Relative to surface saturation concentration (Fig. 9 (b)), the highest observed oxygen saturation during the expedition was 95%. There was a general decline over the cruise period,

with an increase in oxygen after periods of high wind, as expected. The measured oxygen concentration in the top ten metres was very uniform, with a maximum standard deviation of 0.1%, indicating that the surface ocean was well-mixed with respect to oxygen over the timescale of a day during the CTD casts. However, this data has coarse temporal and spatial resolution and so could not capture any local patches of higher relative gas saturation which might be associated with the top metre of the water column or the observed bubble plumes.




We note that the deep plumes themselves could not be causing significant patches of higher gas saturation in their own local water mass. If all the oxygen contained in bubbles making up an air void fraction of $10^{-5}$ dissolved into its local water mass, it would only increase the local saturation state by approximately 0.1%. However, the very high void fractions ($10^{-3}$ - $10^{-1}$) just after a wave breaks could significantly increase local supersaturation beneath a breaking wave. We suggest that if there is a

region of supersaturated water surrounding a plume, it is due to aerated water in the shallow surface layer being advected downwards with the plume. High local gas saturation may then also increase the lifetime of the bubbles carried downwards. In this case, the bubbles in a deep plume could only make a very small contribution to oxygen flux downwards (for example), but they would be held within a water mass carrying gases from the surface and so the bubbles could act as a tracer for gas-rich water.


## 2.5 Limitations

Our results have several limitations. The presence of surfactants is completely ignored here, since we made no direct measurements and the nature of the surface microlayer in wind conditions above 20 m s$^{-1}$ is unknown (Wurl et al., 2011; Sabbaghzadeh et al., 2017). Three deployments were in water of approximately 8°C and one at 20°C, and other environmental

conditions varied between deployments, so we cannot separate any potential temperature effects from other parameters. Finally, in the discussion that follows we take no account of the directional wind and swell data (or possible interaction between wind and swell), using only total wind speed and the wind-wave Reynolds number to group data points. This is due to the small amount of data when compared with the large number of varying parameters; our four deployments covered a very small subset of the possible combinations and so it is not possible to draw conclusions about swell effects.


## 3. Discussion

### 3.1 Comparison with previous measurements

There are relatively few measurements linking bubble sizes with depth. Terrill (2001) found no bubbles greater than 600 μm at a depth of 0.7 m and a wind speed of 15 m s$^{-1}$. Norris et al. (2013) found a similar upper limit of 570 μm at 0.4 m and 14 m

s$^{-1}$ winds. Randolph et al (2014) made deeper measurements, at 6–9 m, under winds up to 13 m s$^{-1}$, and no bubbles bigger than 60 μm. Vagle et al (2010) parametrised bubble size distributions at different depths measured at Ocean Station Papa, finding that the shape of the volume scaled distributions averaged over a three-week period (in wind speeds up to 20 m s$^{-1}$) could be fitted by a function of depth and bubble radius. Our finding that bubbles larger than 300 μm were very rare at 2 m, and none larger than 180 μm at 4 m were seen at wind speeds up to 20 m s$^{-1}$, fit well with these previous measurements. As discussed

in Sect. 2.1, these maximum radii do not have a strong wind speed dependency and appear to be too low for the limiting factor to be the balance between buoyancy and turbulent flows. It seems likely that the limits are due to the processes that bubbles undergo while they are still within the top metre or so of the ocean (even in the heaviest seas), and further study is required to identify those limiting mechanisms. We identify two possible alternatives. The first is a process that alters bubble size as they





age, perhaps a short period of dissolution until a limiting size distribution is reached. The second is a selective advection
process, perhaps due to advection being limited to bubbles that reach depths greater than a few tens of centimetres just after
the wave first breaks. It is also possible that bubble production mechanisms may be directly responsible for the size distribution
of the smallest bubbles.  The buoyancy processes provide an absolute limit, but in practice it seems that partial dissolution of
bubbles may happen relatively quickly, forming a relatively stable plume made of bubbles which do not undergo further
significant size changes. Surface measurements of the initial bubble size distribution (Deane, 2002) suggest that bubbles are
produced at all sizes between 100 μm and a few millimetres in radius, and other lab studies have observed bubbles down to 50
μm in radius (Deike et al., 2016).  The question of whether the bubbles in the longer-lasting population have maintained their
original size and survived advection and buoyancy processes, or whether they started as larger bubbles and underwent partial
dissolution is open.

Trevorrow (2003) observed that in deep water at Ocean Station Papa, there was a striking relationship between bubble plume
depth (observed to be up to 25 metres in their study, which used a 200 kHz inverted echo sounder, resonant with a 17 μm
bubble) and e-folding depth. The deeper the plume, the greater the e-folding depth, implying that the same bubbles are spread
out over a greater depth range, making the bubble distribution more uniform with depth. They suggested that this was consistent
with turbulence, convection, and Langmuir circulation advecting bubbles to form deep plumes. Vagle et al. (2010) also suggest
that Langmuir circulation is the dominant mechanism responsible for transporting bubble plumes down into the mixed layer.
The expectation is that this rapid downward motion occurs when a shallow bubble plume is advected across the top of a
Langmuir cell and reaches the downward leg of the flow. Chiba (2010) suggests that deeper plumes caused by Langmuir
circulation could be particularly significant for ocean oxygen uptake. Our results suggest that it is the Langmuir circulation
carrying oxygenated water downwards, rather than the deep bubble plumes themselves, which are important. If this
interpretation is correct, future research priority should be given to the spatial variation of oxygen saturation close to the ocean
surface on scales of a few metres, and the ways in which shallow bubble populations may drive gas uptake.  We also observe
a severe reduction in void fraction between 2 m and 4 m depth, which implies that plumes deeper than 4 m will have lower
void fractions still. The very deep plumes (>4 m) would be very obvious on sonar images, because the smallest bubbles
approach the resonant frequency of the sonar, but would have a minimal influence on gas transfer processes.


### 3.2 Processes

Langmuir circulation is a critical process in the interpretation of our results but we have no direct measures of the surface flow
field. Chiba and Baschek (Chiba and Baschek, 2010) suggests that for wind speeds of 20 m s$^{-1}$, the separation between
Langmuir cells is likely to be about 12 metres.  However, there is a lag in the cells responding to the instantaneous wind, and
the buoy was being blown downwind. We cannot be sure about the position of the buoy relative to the surrounding circulation



patterns. It is also challenging to identify clear periods of downward flows which might correspond to the downward leg of a Langmuir cell pattern, because of the complexity of the buoy movement with respect to the local surface.

The signature of Langmuir cell formation is the accumulation of long foam patch streaks approximately parallel to the wind. Surface bubbles accumulate because this is a convergence zone and the bubbles in foam patches will not be advected downwards. However, there has previously been little evidence to address the processes generating the regions we have identified as "deep plumes": regions with a void fraction above $10^{-6}$ at a depth of 2 m and extending for several metres horizontally. There are two possible mechanisms:

i)    A distinctive bubble size distribution arises in the minutes after a wave breaks, and it is advected sideways as a coherent patch which may reach a convergence zone and be pulled downwards. In this case, the distinction between the moving shallow patch and a "deep plume" is that they are different stages of the same water mass and contain very similar bubble size distributions.

    ii)    The distinctive bubble size distribution is the result of bubbles accumulating at the convergence zone, and the
450         constant shape of that bubble size distribution represents an averaging across all the heterogeneous patches of bubbles which are advected just beneath the ocean surface until they are trapped in a convergence zone.

The strong variation of the maximum bubble size with void fraction is more consistent with the first case, the bubble conveyer belt, because in the accumulation case the maximum bubbles sizes from different coherent near-surface patches would mix
together. The bubble size distributions are also more consistent with the first case for most bubbles. Figure 10 shows the averaged bubble size distributions separated by void fraction. For void fractions between $10^{-6}$ and $10^{-4.5}$, there is a small increase on the general trend for bubbles greater than 300 μm. The rise speed for a 300 μm coated bubble is expected to be 0.08-0.09 m s$^{-1}$ (Deane et al., 2013), which is towards the high end of the downward flows measured by the ADV (a 100 μm coated bubble is expected to rise at 0.01 m s$^{-1}$). As previously noted, bubbles of this size were rare. This suggests that if they do reach
a depth of 2 m, these large bubbles can remain in shallower water for longer. However, the majority of the bubbles are not big enough to rise significantly against the downward flow speeds observed, and so will be carried downward until they are destroyed.

The first case is also consistent with our observation of a smooth probability distribution for void fraction which depends on the environmental conditions (Czerski et al., 2021). Presumably, the regions which have intermediate void fractions ($10^{-8} – 10^{-6}$)
at 2 m are positioned between deep plumes and contain long-lasting bubbles that were pulled downwards by previous advection patterns, and possibly also bubbles mixed downwards gradually by turbulence. At 4 m, it seems likely that bubbles > 20 μm radius are only found in association with concurrent downward flows, and cannot last long enough to form a background population


Some ambiguity remains: the "deep plume" regions are large, with a horizontal extent of several metres, and it is not clear that a single breaking wave could generate enough small bubbles to fill this observed bubbly region. Previous sonar observations (Zedel and Farmer, 1991) show extensive bubbly regions filling the downward leg of Langmuir cells, but these observations could be due to relatively low numbers of very small bubbles which were resonant with the sonar rather than the higher void
fractions including larger bubbles that we see here.

Our data are more consistent with the first explanation, except for bubbles larger than 300 μm in radius. In this case, the convergence zones will always contain bubbles but will have highly heterogeneous void fractions and size distributions, and identification of a "plume" is ambiguous because the heterogeneity of the bubbles in the convergence zone just represents the
heterogeneity of bubbles in the shallow populations.

The consistent large difference in void fraction between 2 m and 4 m suggests that bubbles move between the two depths in the downward direction only, due to coherent flows rather than turbulent mixing. It may be that the lower void fraction at 4 m represents only the lower probability of bubbles being carried down to those depths without destruction rather than a difference
in the processes happening at that depth.

Our data support the idea that there are two regimes of bubble behaviour. In the first, at higher void fractions (above $10^{-6}$), bubbles are effectively stable and do not dissolve significantly. The void fraction is reduced as they mix with surrounding water but with minimal change to their size distribution. This is the flatter slope seen at higher void fractions in Figs. 7 and 8.
These high void fraction regions may be contained within locally saturated water which preserves the bubble population. In the second regime, with void fractions below $10^{-6}$ at 2 m, the bubble size distributions are far more heterogeneous. Bubbles may be dissolving or have a strongly radius-dependent destruction probability, and they follow the steep slope seen at the left hand side of Figs. 7 and 8.


### 3.3 Anatomy of a plume

We set out here our current understanding of each stage of bubble existence: formation, changes and movement due to buoyancy, advection and dissolution, and finally destruction. The proposed stages are summarised in Fig. 11.

**(i) Bubble Formation**

The initial population of bubbles formed by a breaking wave evolves quickly in the first few seconds, with bubble fragmentation and Messler entrainment creating an initial size distribution as described by Deane and Stokes (Deane and Stokes, 2002) which then evolves further as buoyancy removes the largest bubbles. Once turbulent fragmentation under the active breaking wave has ceased, no new bubbles are created since fully submerged bubbles will neither fragment nor coalesce.



There is no consensus on the size of the smallest bubbles created, and work is ongoing to understand the short-lived population of large bubbles.

**(ii) Shallow bubble layer evolution.**

In the near-surface layer (which has an unknown depth, but is thought to be of the order of 1 m) a highly heterogeneous but
statistically stable bubble population develops, which is significantly different from the population present immediately after a wave breaks. It has the shapes shown in Fig. 3 and is continuously fed by new breaking waves. We suggest that there is an unknown series of processes in the top metre or so of the ocean which convert the highly unstable initial population with void fractions $\sim 10^{-1}$ into a pseudo-stable size distribution which can persist for at least several minutes, and which has a maximum void fraction limit of $10^{-4.5}$. This may occur over many minutes as bubbles partially dissolve, are lost from the population as
they rise under buoyancy, or may mix and advect while remaining close to the ocean surface. Some may dissolve completely, and some may collapse. It is likely that all open ocean bubbles will be completely coated with surfactants and particulates which will stabilise the population so that bubbles could have a lifetime of many minutes even when the surrounding water is undersaturated. The size distribution of this quasi-stable population may be determined by buoyancy, gas saturation, temperature, surfactant presence, and turbulent mixing. Our sonar data (Czerski et al., 2021) show these shallow populations
remaining in the top metre for most breaking waves. Since bubbles greater than 220 μm radius were rarely observed at 2 m, even at wind speeds of 27 m s$^{-1}$, breaking processes alone cannot drive bubbles to this depth. This shallow population is continually reformed as more waves break while patches of quiescent bubbles from previous breaking waves drift freely until they are advected downward or destroyed close to the ocean surface. The bubble size distributions we observe in the high void fraction regions are upper limits, but most of the space in between appears to be filled with a far more irregular bubble
population with a lower void fraction. There is no evidence to constrain the length of time a bubble could remain in this upper layer. Our sonar data show that there can be a significant gap in time, at least tens of seconds and possibly several minutes, after a visible breaking wave and before deep plume formation.

The existence of a near-surface bubble layer with a complex structure has been discussed previously in the context of acoustic
propagation (Norton et al., 1998) and Dahl et al (Dahl et al., 2008) suggested that it has a thickness of O(1m). It was termed the "persistent surface bubble layer" by Deane et al (Deane et al., 2013). There is no direct evidence to address whether this pseudo-stable population feeds whitecaps while it is in the top metre of the ocean. Once the bubble population has stabilised, even while it is still in the top metre, it may be decoupled from the surface.

**(iii) Advection downwards and deep plume formation**

As suggested by Zedel & Farmer (Zedel and Farmer, 1991) and our own ADV data (Czerski et al., 2021), the downward limb of Langmuir circulation pulls surface water downwards and any bubbles in that water mass will be carried with it, possibly to depths of a few metres. We observed downward speeds of 0.05-0.10 m s$^{-1}$ associated with some deep bubble plumes, implying





timescales of 20-40 seconds for bubbles to be carried from 2m to 4m. The fact that bubble plumes appear to remain intact
strongly suggests that turbulence plays a negligible role in this process and that the downward movement is due to coherent
flows.

We have no data which constrains the proportion of bubbles in the shallow layer which are eventually advected downwards at
a Langmuir convergence zone. The most critical parameters for that process are the lifetime of bubbles in the top metre and
the probability of any given patch of water being advected downwards within that lifetime. This downward advection process
happens over tens of seconds, forming a "deep plume" extending to a depth that depends on the conditions. Only the smallest
bubbles are advected to 4 m. Our bubble size distribution data do not show any shift in the radius of peak volume that would
support the idea of buoyancy sorting the bubble population at these depths. The higher downward velocity in the centre of the
downward limb of a Langmuir cell could trap larger bubbles than the edges, but we see limited evidence for this.  Deep plume
e-folding depths represent a combination of how much the initial bubble population is stretched downwards and the variation
in destruction probability with depth.

One consequence of the separate mechanism for downward advection is that the age of bubbles when they are transported
downward to form deep plumes may vary considerably. It could presumably happen immediately if a wave breaks at the
downward limb of the Langmuir circulation, but there could also be a significant delay. It seems clear that once bubbles are
advected to 2 m (and the depth limit could be even shallower), the gas they contain will eventually all be dissolved into the
ocean. None of our evidence supports the idea that bubbles might rise from these deep plumes to return to the ocean surface.

There may be significant supersaturation of the water in the top metre or so even though the deeper water is undersaturated,
and this would contribute to the longevity of the shallow bubble population. If a shallow population of bubbles is advected
downwards to form a deep plume, it may be carried within an oxygenated water mass. Although these bubbles will all be
injected into the ocean, we suggest that the bubbles in the stabilised population which form the deeper plumes do not have a
significant influence on the oxygen content of their host water mass. Their lifetime may be affected by the local gas saturation,
but the gas they contain is insignificant compared with that dissolved in water advected from the surface.  Many authors have
considered the asymmetry of the gas flux process at the ocean surface, and particularly the rapidly increasing pressure with
depth that may force bubbles to dissolve even if the gases are super-saturated. Our results suggest that there are too few bubbles
in deep plumes to make a significant difference to the gas content of the water within the plume, and that any significant
additional dissolution due to pressure would have to occur in the top metre or so where there is a relatively limited excess
pressure.
**iv) Processes within the deep plume**



The bubble populations that reach a depth of 2 m have a void fraction limit of $10^{-4.5}$, with a sharp cut-off in the distributions at this value. For void fractions greater than $10^{-6}$, the bubble population is relatively stable and may be described by two power laws: a slope of -0.3 for bubbles less than 80 μm in radius and a slope of -4.4 for larger bubbles. When normalised by void fraction, this forms a tight distribution for all void fractions above $10^{-6}$. It seems likely that advection downward will stretch out the shallow surface plumes, and may be associated with additional mixing which will cause the fixed number of bubbles in the plume to be distributed throughout a larger volume, reducing void fraction. As this happens, the associated water mixing may also change the gas saturation state of the water around the bubbles. At increased depths the external pressure will cause the inwards force on the bubble surfaces to increase and the actual saturation state of the water around them to decrease relative to its initial saturation state.

At 4 m depth the ocean is not filled with small bubbles (above 5 μm radius, the detection limit of the resonator) even at high wind speeds, so either the bubbles at this depth are produced in large numbers and have a short lifetime, or they are produced slowly but have a long lifetime. There are many regions at 4 m depth even during very high winds when the bubble void fraction is well below $10^{-8}$, which suggests that high production rates and relatively short lifetimes are more likely.

**(v) Deep plume evolution and bubble collapse**

The ADV data on downward flows suggests that bubbles may be able to move from 2 to 4 metres in 20-40 seconds, during which time the void fraction decreases by a factor of between 15 and 450 in individual plumes, with a mean of 85. The implication is that the vast majority of bubbles are destroyed relatively quickly once they move below 2 m depth. Our data suggest that the bubbles would have existed at 2 m or above for several minutes before moving downwards, and the rapid disappearance requires the presence of a destruction mechanism that is strongly dependent on pressure. Figure 5 shows that many of the remaining bubbles have a similar size to the original bubbles, ruling out slow dissolution as the dominant destruction mechanism. We suggest that bubbles advected to greater depths may collapse suddenly with a half-life that depends on their coating, the water saturation state and the surrounding pressure.

A stochastic process of bubble collapse (as observed in the lab by Johnson & Cooke (Johnson and Cooke, 1981)) can destroy bubbles without further change in radius. We note that the increased pressure at even modest depths will decrease the saturation state of the water surrounding the bubbles, and so all the bubbles we observed at 2 m depth were likely to be in undersaturated water relative to their own gas pressure. It is possible that the void fraction e-folding depths may partly reflect the probability of sudden bubble collapse, which increases with pressure and is influenced by the bubble coating and maybe other environmental conditions.




## 4. Conclusions

We have presented detailed measurements of bubble size distributions during high wind conditions in the North Atlantic. The level of detail provides new insights into the mechanisms that create and maintain bubble populations after waves break. Our
data confirms the suggestion by Zedel & Farmer (1991) that bubble plumes of several metres depth are formed when coherent circulations advect bubbly surface water downward and that these deep plumes are not directly connected to breaking waves.

We identify two separate populations of bubbles, although we could only directly observe one of them. The first is the shallow population, confined to a near surface layer (approximately 1 m deep) which is formed as a wave breaks and which evolves
through dissolution and buoyancy to form a statistically stable population that remains close to the surface. The second is the deeper plume formed only when part of the shallow bubble population is advected downwards by Langmuir circulation or other coherent flow. We identify a pseudo-stable bubble size distribution at 2 m when the void fraction is above $10^{-6}$, described by a power law with a slope of -0.3 for bubbles smaller than 80 μm in radius and -4.4 for larger bubbles. This population is not altered by dissolution or buoyancy, but it slowly spreads out in space, reducing the void fraction but not the bubble size
distribution. For lower void fractions, the bubble size distribution is far more variable at a depth of 2 m and we infer that these intermediate void fractions are associated with the regions between the downward advection zones. During wind speeds up to 27 m s$^{-1}$, bubble measurements at 2 m showed that there is a very strong cut-off in bubble size at 220 μm in radius, a far lower size limit than that suggested by buoyancy and turbulence arguments alone. This suggests that the shallow plumes may undergo considerable evolution before they are pulled down to 2 m and below, and that this cut-off is controlled by processes very
close to the ocean surface. We also note a sharp cut-off in the void fractions that are possible at 2 m depth of $10^{-4.5}$, even in the highest winds, which is also likely to be controlled by processes in the surface layer.

Our data suggest that the major destruction mechanism of the deep plume bubbles may be sudden collapse, a stochastic process with a probability that is strongly dependent on pressure (and therefore depth). Bubbles are far more short-lived at 4 m depth
than at 2 m, and are rapidly destroyed as they travel downwards. Once formed, the deep bubble plumes are completely decoupled from the surface. We have observed that bubbles around 80 μm in radius can persist at 2 m for 4-6 hours after the cessation of wave breaking, after a period of very high and sustained winds. This suggests that the local gas saturation state, as well as the bubble coating, has a strong influence on bubble longevity. This is consistent with results in the companion paper (Czerski et al., 2021) suggesting that deep plume lifetimes can be of the order of at least tens of minutes. There is a strong case
for co-located high time-resolution measurements of bubble presence, flow structures and gas saturation states in future studies.

The availability of high time resolution measurements in this study has highlighted a potential artefact in measures of bubble size distributions averaged over long time periods. Our results show that when evaluating models for the processes driving





bubble size distributions it is essential to consider instantaneous measurements rather than rely on averages over long periods

of time.

## 4.1 Open questions

One of the reasons that it has been hard to correctly model gas uptake in high wind conditions is the lack of in situ observational

data to constrain the models and our physical understanding of bubble plume behaviour. We have been able to constrain some

critical parameters of bubble plumes and their evolution, but many questions remain. We identify the following important

questions for future studies:

i.   What are the bubble populations in the top metre of the ocean, and how are they influenced by environmental conditions?

650        A better understanding of the mechanisms responsible for the shift from the initial bubble populations observed by Deane

(2002) to such a statistically stable background population in the shallow layer before bubbles are advected downwards is

needed. Why is there such a sharp cut-off in void fraction at $10^{-4.5}$ in the bubbles that reach 2 m depth and what is the

mechanism causing the slope break observed in our observed deep plume bubble size distribution?

ii.  What proportion of the shallow populations created by breaking waves (and the gas-rich water packets they are presumably

carried in) are advected downwards at convergence zones? This is critical to link bubble production processes at the surface

with gas export downwards.

iii. What are the timescales of advection at convergence zones and how long will deep plumes last? This is significant because

660        it may allow measurements of deep plume presence to be converted to estimates of bubble and gas flux.

iv.  What are the mechanisms of bubble destruction, both in the shallow populations and the deep plumes? A related question

is when and where significant dissolution (as opposed to destruction by collapse) occurs. This is needed to link bubble

measurements with gas fluxes.


v.   What is the spatial distribution of gas saturation in the upper ocean during high wind events? Since the shallow bubble

populations are known to be highly heterogeneous, it seems likely that the surface distribution of dissolved gases is also

highly variable in time and space. We have suggested here that there may be high local saturations of oxygen and nitrogen

associated with shallow bubble plumes and carried downwards with the deep plumes, but these have not been measured

670        directly. Our work suggests that it is essential to acquire high resolution (in both space and time) gas saturation

measurements co-located with bubble measurements.



vi. What is the nature and influence of the bubble coating? The coating is likely to influence bubble stability and gas exchange across its surface, especially in the top metre.


viii. How is the uptake of various gases partitioned between the deep plumes and the shallower but higher void fraction populations near the surface? Is it the case (as we suggest here) that the coherent flows are transferring oxygen directly from the near surface to the deep ocean and that the bubbles in the deep plumes do not make a significant contribution to the total gas injection?


**Appendix A: Artefacts due to time-averaging of bubble size distribution measurements**

Few bubble size distribution studies have had the advantage of 1 Hz data for many hours, and most bubble size distribution data in the literature is presented as averaged distributions over long time periods (Vagle et al., 2010). Analysis of our high

time resolution (1 Hz) data reveals that time averaging can cause a systematic bias in the bubble spectra when combined with a finite sample volume. Figure A1(a) shows all the individual one second bubble size distributions for both camera and resonator for one 45-minute period. The individual distributions have a consistent shape (two straight lines with a slope break around 80 μm radius), though they span over two orders of magnitude in concentration. Figure A1(b) shows the average over the entire period. The averaged distribution steepens towards the right hand side and curves downwards for bubbles > 200 μm,

in contrast to the straight slope seen in the 1 Hz distributions. This downward curve is an artefact caused by averaging measurements made in a finite sample volume over periods which include a wide range of void fractions, and we explore that here.

Consider a perfect theoretical distribution normalised by void fraction described by two power-law slopes with no end, as

shown in figure A2. It can have any void fraction but the shape remains constant; the distribution simply moves vertically along the y-axis as the void fraction changes. Now consider a practical measurement of this distribution. Each measured distribution cuts off because the number of large bubbles inside the sample volume drops below one. If the population is changing rapidly and it is not possible to average over, for example, 1000 separate measurements of the same bubble population, then the measurement cannot distinguish between a 0.1 chance of a bubble being present and a 0.001 chance. When

void fractions are low, very few large bubbles are measured at all in practice, even if there is a non-zero chance of them being present. In a time-averaged distribution which includes some high void fractions and large bubbles, the small number of large bubbles sampled during those short periods are averaged over the entire time period, and there are not enough samples of those large bubbles to accurately reflect the small probability of their presence. This means that the average counted number of large bubbles is lower than the actual number present at any point because a disproportionately large number of the actual

measurements are zero. The averaged distribution therefore shows an artificial steepening towards larger radii which is not seen at any point in the real world, generating a bias. It is possible that past measurements of slope steepening are partly due



to this effect — the bubble size distribution stays largely the same with time, but the zeros in the measured distribution result in an increasing low bias with bubble size at low void fractions.

We recreate this artefact in figure A2. A "perfect" bubble size distribution was assumed (circular symbols), with a slope of -

0.4 for bubbles smaller than 81 μm and -4 for bubbles larger than 81 μm. This covers a size range from 10–1000 μm in radius. This bubble size distribution matches the typical individual bubble size distributions shown in figure A1. To simulate the bubble size distribution at each second, this "perfect" distribution was scaled to reproduce the measured void fraction for each second during the 45 minute period shown in figure A1. This produced the expected number of bubbles within the sample volume of the camera for every second, which was frequently less than one. Then for every individual distribution, all fractional

values below 0.5 were converted to zeros (following the assumption that these bubbles had a very low probability of being detected in practice), and the entire simulated time series was then averaged in time. Figure A2 shows that the simulated result matches the measured averaged size distribution for this period extremely well. The largest void fractions (Fig. 1(b)) were only present for very short periods and so were unlikely to be sampled enough to catch the rare large bubbles in a representative way. However, our data show that for these persistent bubbles, the *actual one second bubble size distributions have a straight*

*slope* above the slope break without the drop-off seen in averaged data. This is important for understanding the physical processes responsible for the observed bubble size distributions.

We note here that slope steepening at the larger radii is also expected if bubbles rise out of the plume through buoyancy. We do not suggest that this process is unimportant, but it is happening at a different point in time (immediately after a wave breaks) and over a different time period (seconds rather than minutes).


This artefact is an important consideration for any models seeking to reproduce the physics of bubble advection, dissolution and collapse. As shown in the main text, any individual bubble size distributions for void fractions greater than $10^{-6}$ are fitted very well with two straight lines in log-space with a slope break around 80 μm radius. We recommend that time averages of continuous data are not used when presenting the data from future studies of bubble size distributions of long-lasting plumes.


**Appendix B**

Figure B1 shows the percentage of data points for each one second bubble size distribution that fit in between the black lines shown in Figure 3 (which represent a halving and doubling from the mean distribution, as described in Sect. 2.2). Above a void fraction of $10^{-6}$, a very high proportion of all one second bubble size distributions (which are individually expected to be

noisy) fit within these limits.

**Appendix C**


Figure C1 shows typical volume distributions and Gaussian fits for both camera (a) and resonator (b). The volume associated with each size bin is calculated by multiplying the number of bubbles counted at each size by the volume of a single bubble at the central radius of that bin. Gaussian fits were calculated for every 1-s distribution using a least squares optimisation routine in Matlab. The fitted peak position can take any value, but we note that the smallest bubble radius measured was 20 μm for the camera and 5 μm for the resonator, so no fit will have a peak below those values. The fitting process provides better radius

resolution than relying on the bin size responsible for the largest volume fraction. For the resonator, only distributions with a void fraction higher than $2 \times 10^{-8}$ (the noise limit) were fitted, but all camera distributions were fitted. At lower void fractions, the data is less smooth but a peak can usually be identified.

Figure C2 shows the ten second averages of the radius at peak volume, aggregated across all deployments and colour-coded

by Hs, U10 and Re_ww. In general, the patterns of the individual deployments are still visible suggesting that local water conditions have a stronger influence than weather or wave state. The correlation with wind speed shows the least influence of the deployments, which is consistent with a controlling parameter that is dominated by wind speed on average. It seems likely that local gas saturation is the parameter with the strongest influence on this pattern, but our gas saturation measurements did not reflect the water conditions at the buoy and so the data we present here is inconclusive.


**Data Availability**

All data are archived with the British Oceanographic Data Centre (BODC). The bubble data are available at doi: 10.5285/c972e316-2b93-1b4e-e053-6c86abc02285 and the wave data can be found at doi: 10.5285/c9ae04d6-32d2-73f1-e053-6c86abc0c833 Other HiWinGS cruise data, including the near-surface meteorology used here are available from:

ftp1.esrl.noaa.gov/psd3/cruises/HIWINGS_2013/Collective_Archive.

**Author Contribution**

HC and IMB designed the bubble and wave state section of the HiWINGS project. BB was the Chief Scientist of HiWINGS, and oversaw all operations. RP designed the spar buoy and was responsible for its operations at sea. SG designed and built the

bubble camera and analysis software. AM helped with deployment at sea and carried out the sonar analysis. HC was responsible for the bubble sensor deployment and carried out analysis of all bubble data except the sonar. HC and IMB prepared the manuscript.

**Competing Interests**

The authors declare that they have no conflict of interest.



**Acknowledgements**

This work was funded by the Natural Environment Research Council under grants NE/J022373/1 (HC) and NE/J020893/1 (IMB), NE/J020540/1 (NOC) and HC's fellowship NE/H016856/1. Funding for ship time was provided under US National Science Foundation grant AGS 1036062. We are grateful to the captain and crew of the R/V *Knorr* for their invaluable assistance at sea. We gratefully acknowledge the contribution of Nick Hall-Patch & Svein Vagle for their work on the construction and operation of the acoustical resonators used. We also acknowledge the contribution of Raied Al-Lashi, who
ran image analysis on bubble camera data from two deployments.

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

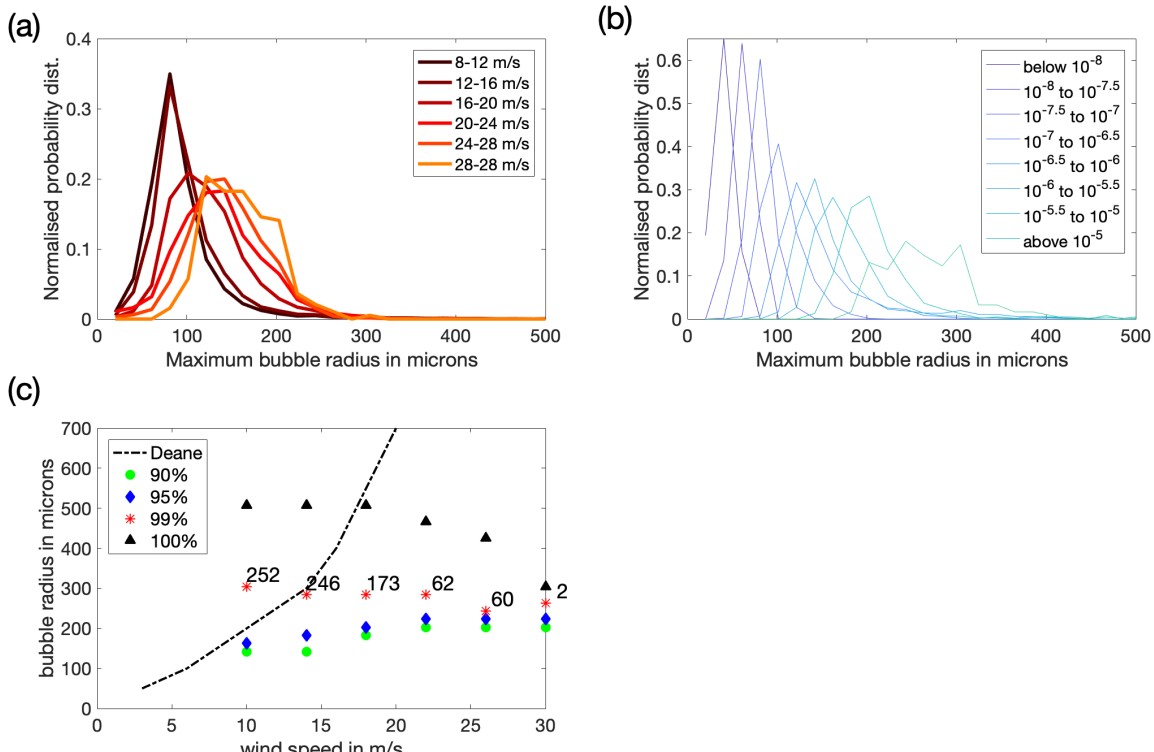

**Figure 1: Normalised probability distributions of the maximum bubble radius in each one-second distribution at 2m depth, segregated by wind speed. (b) Normalised probability distributions of the maximum bubble radius seen in each 1 second distribution at 2 m depth, segregated by void fraction. Note that there are only 123 one-second measurements where the void fraction seen was above $10^{-5}$. (c) The $90^{th}$, $95^{th}$, $99^{th}$ and $100^{th}$ percentiles of the probability distribution of maximum bubble sizes in each wind speed bin. The dashed line shows the escape radius prediction of Deane et al. (2013) at 2 m depth. The number of photographs making up 1% of the distribution at each wind speed range is labelled next to the $99^{th}$ percentile data points..**







**905** **Figure 2: All one second bubble size distributions for every deployment, for both camera (at 2m) and resonator (at 4m). (a) Number of bubbles per micron radius increment per unit volume, dN/dR (m⁻³ µm⁻¹). (b) The same data, but each distribution is normalised by its own void fraction (N/VF). No resonator data was available for station 6. Power law fits for each deployment are shown in the lower set, with the two slopes labelled as S1 and S2.**

.


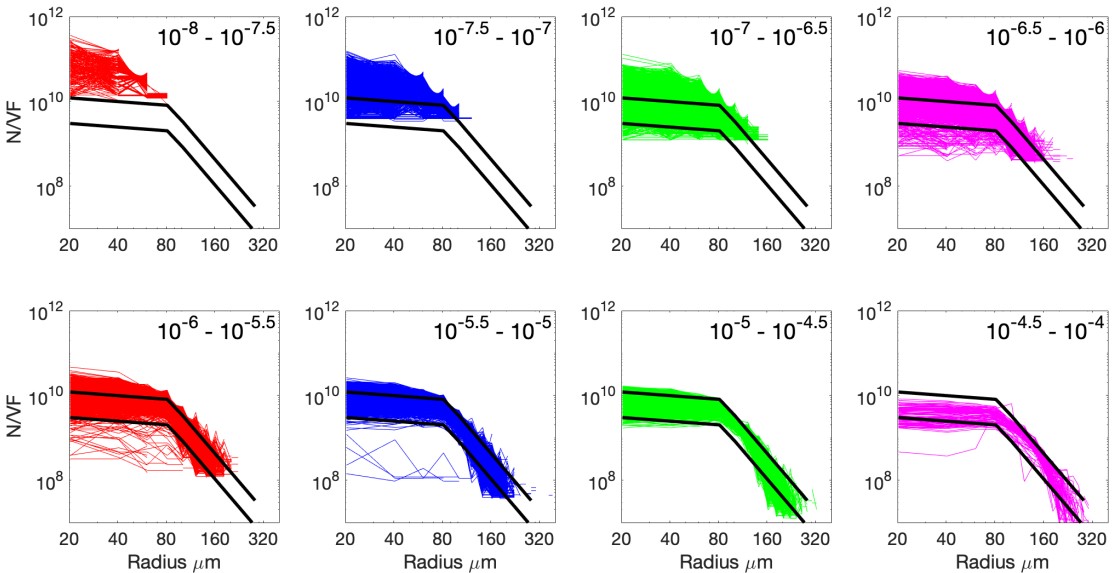

**Figure 3: All camera bubble size distributions at 2m depth segregated by void fraction (shown in the top right of each plot), and**
**normalised by the individual void fraction of each distribution. The y-axis on all plots shows the number of bubbles per micron**
**radius increment per unit volume, divided by void fraction. The black lines are the same on all plots and show the halving and**
**doubling of the representative normalised distribution.**

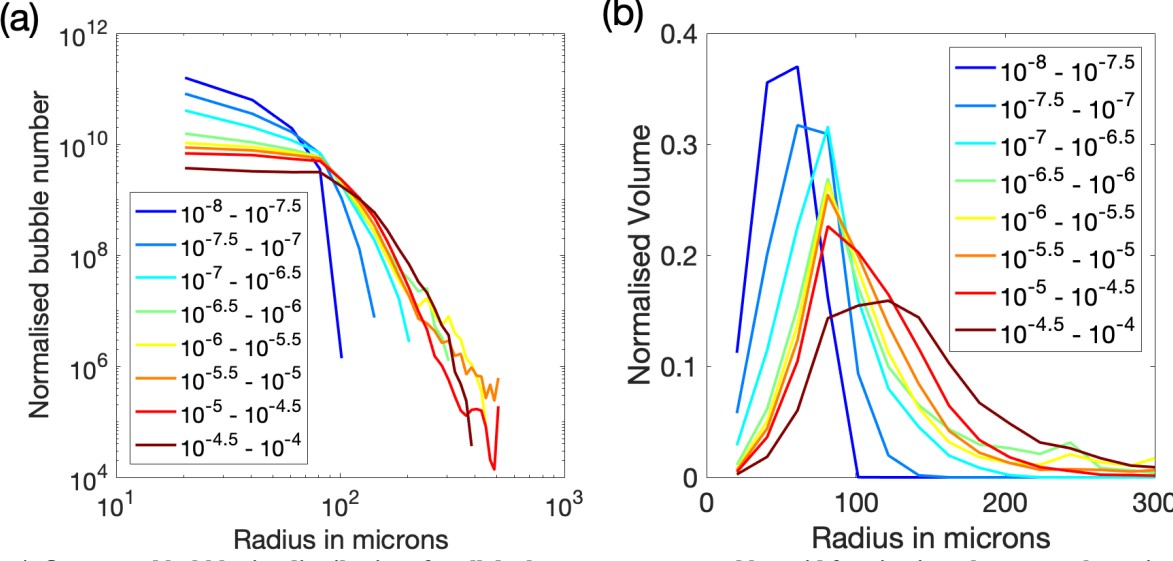

**Figure 4: One second bubble size distributions for all deployments were sorted by void fraction into the ranges shown in the legend.**
**(a) shows the mean bubble size distributions for each void fraction range plotted as bubble number per micron radius increment**



per unit volume, normalised by the individual void fraction. (b) The same data plotted as normalised volume. Each bubble size distribution in a given range was normalised by its own volume and the mean of the resulting distributions is shown here.
        .




**Figure 5: Comparison of fitted radius at the volume peak with void fraction during a 45 minute period on station 6, 2 November,**
**18:00:00 to 18:45:00. (a) fitted peaks at 2 m, with one second fits (grey) and 10 second averages (blue). (b) 1 s void fractions at 2m,**
**(c) fitted peaks at 4 m with one second fits (grey) and 10 second averages (blue), (d) 1s void fractions at 4m. The dashed line on (a)**
**and (c) shows 80    m radius for comparison.**

        .






**Figure 6: The variation in the fitted bubble radius at the volume peak at three wind speeds at 2m depth. Each pair shows void fraction below and peak volume radius above. The grey dots are 1s values and the dark blue are 10s averages. (a) and (b) show data from October 25th starting at 1600, when wind speeds were 25-28 m/s and the mean void fraction was $6.8 \times 10^{-6}$. (c) and (d) show data from November 2nd when wind speeds were 16-18 m/s and the mean void fraction was $1.24 \times 10^{-6}$. (e) and (f) show data from November 1st when the wind speeds were 10-12 m/s and the mean void fraction was $3.7 \times 10^{-7}$.**



.


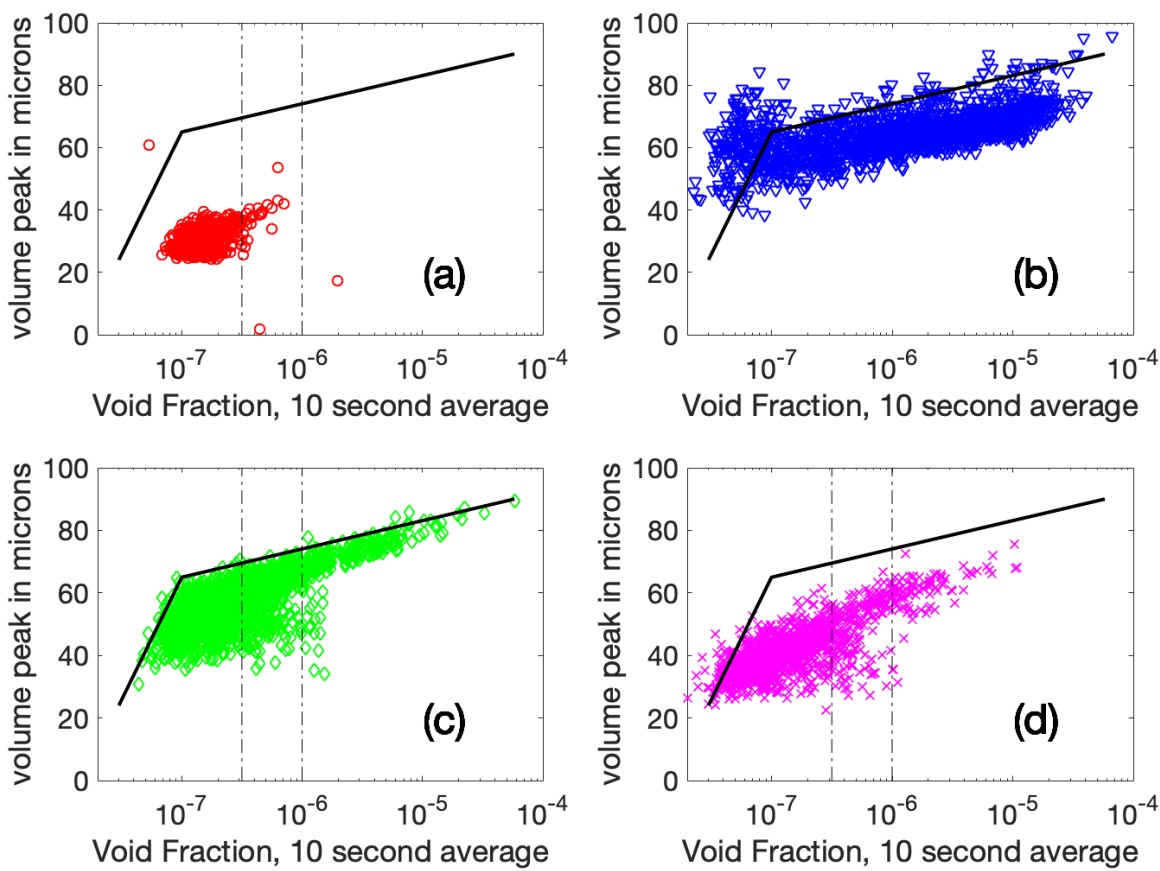

**Figure 7: Scatter plots showing the 10 second averaged void fraction against the peak radius in the volume distribution at 2m depth (a) shows station 3, (b) station 4 , (c ) station 6, and (d) is station 7. The black lines are the same on all plots and follow the envelope**
**of the data in (c). The lines are at 10^(-6.5) and 10^(-6) to allow comparison between plots. We note that the triangles in the top left of plot (b), significantly above the black lines, are all in the eye of the storm: a period of very low winds following very high winds. They appear to be bubbles that are stable for several hours after wave breaking events have ceased. .**




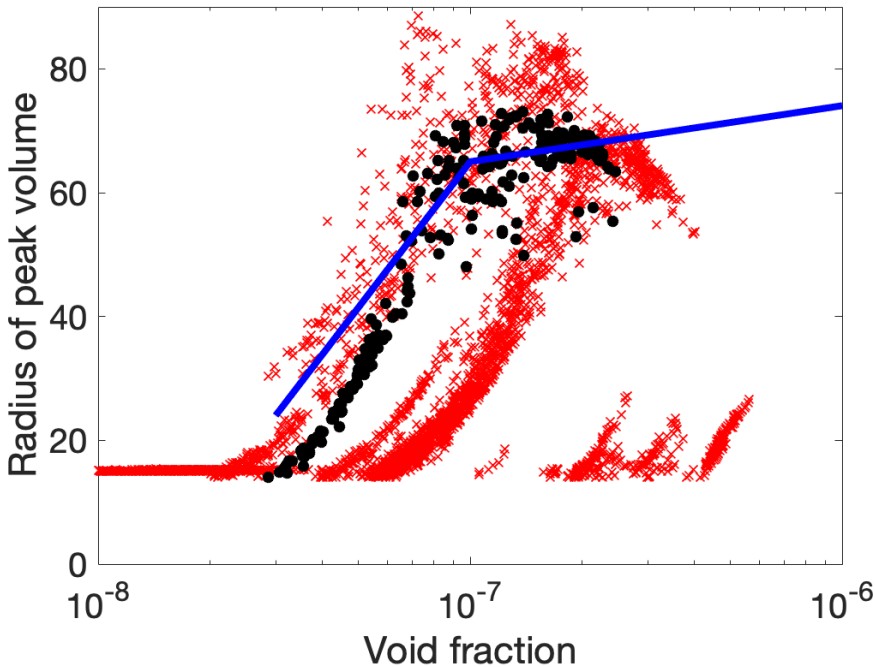

**Figure 8: Comparison of the radius of peak volume with void fraction in the resonator data at 4 m depth. The blue lines are identical to those in figure V. Black dots show data from station 3 (18-21 Oct) (with very low winds) and red crosses show data from station 6 (1-4 November). No data are shown for the later November deployment because there were no successful fits. These are all one second fits, rather than the 10s averages shown above. Each short diagonal line is due to a single plume event.**

.




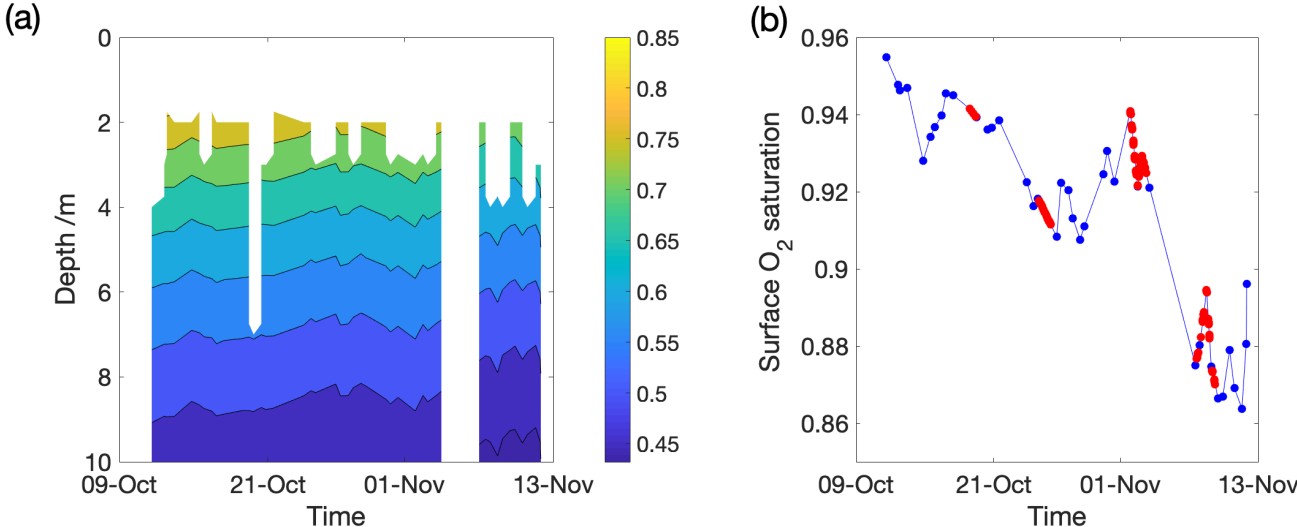


**Figure 9: (a) Oxygen saturation calculated from the CTD casts shown as the relative saturation at each depth rather than the more conventional normalisation to surface saturation. 43 CTD casts were made during the 35 day cruise and the data gap was during the transit to the Gulf Stream. In all cases, the oxygen was well-mixed, so the actual concentration was similar at 10 m and at 2m. The highest depth measurement of each CTD cast varies because the measurements were taken at fixed time intervals rather than**
**at fixed depths. (b) shows the oxygen saturation at the ocean surface estimated from the data shown in (a). Red markers show the four deployments.**

.






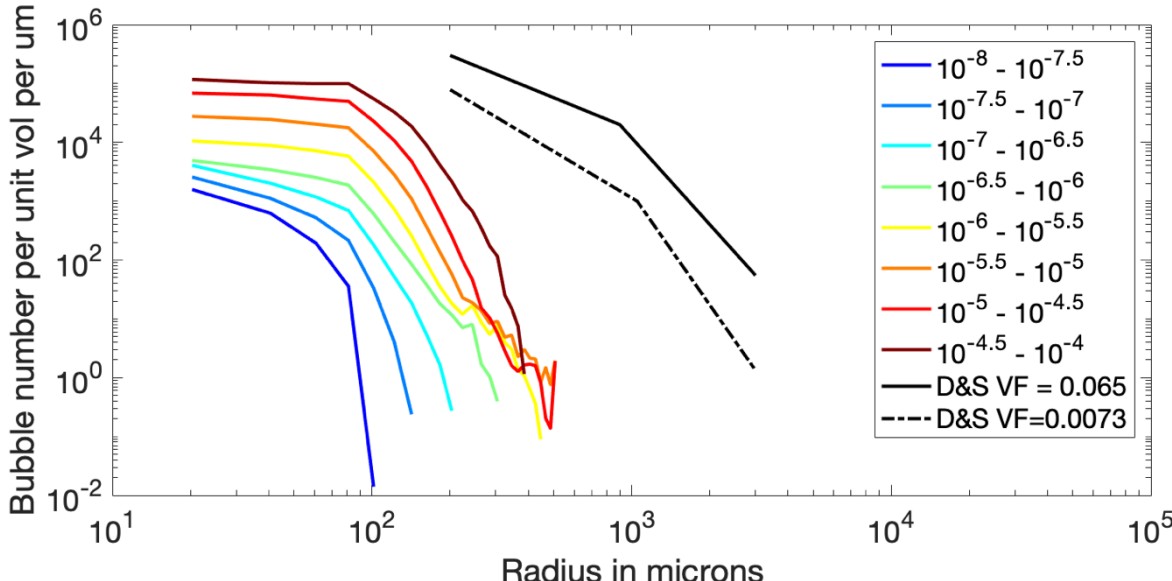

**Figure 10: Average bubble size distributions in each void fraction category across the whole data set (coloured lines), compared with the bubble size distributions observed by Deane & Stokes for void fractions of 0.065 and 0.0073 in the first few seconds after a wave breaks (black solid and dashed lines).**

980   .





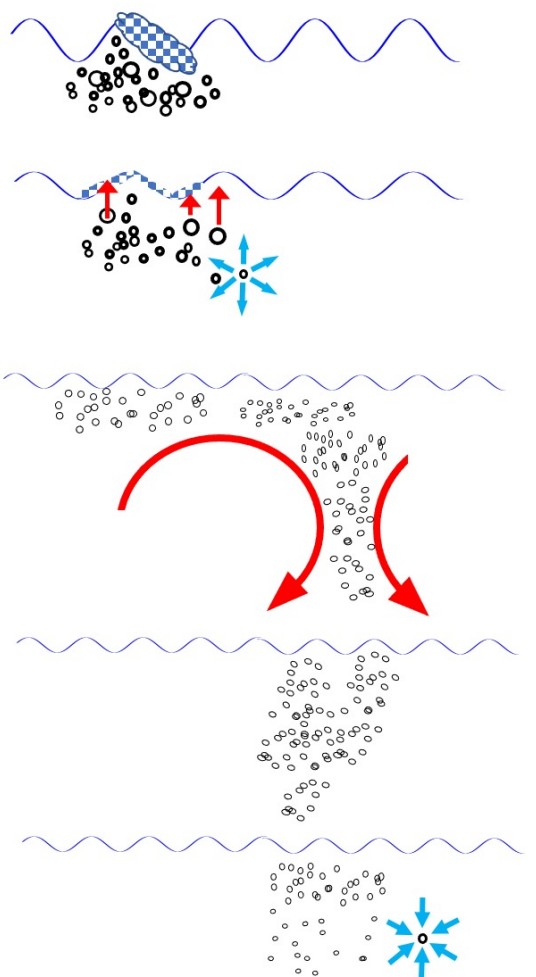

(i) Wave breaks: Bubble formation

(ii) Shallow plume evolution: dissolution, buoyancy, whitecap feeding, production of a quasi-equilibrium population

(iii) Possible deep plume formation: shallow plume drawn downward only if it coincides with descending large scale flow pattern

(iv) Deep plume evolution: Stokes shear, no change to bubble sizes, no buoyancy sorting or dissolution, plume slowly mixes laterally

(v) Bubble destruction: sudden bubble collapse with half-life dependent on pressure, gas injection


**Figure 11: Schematic setting out the major stages of bubble formation, evolution and destruction, as described in the text.**

.






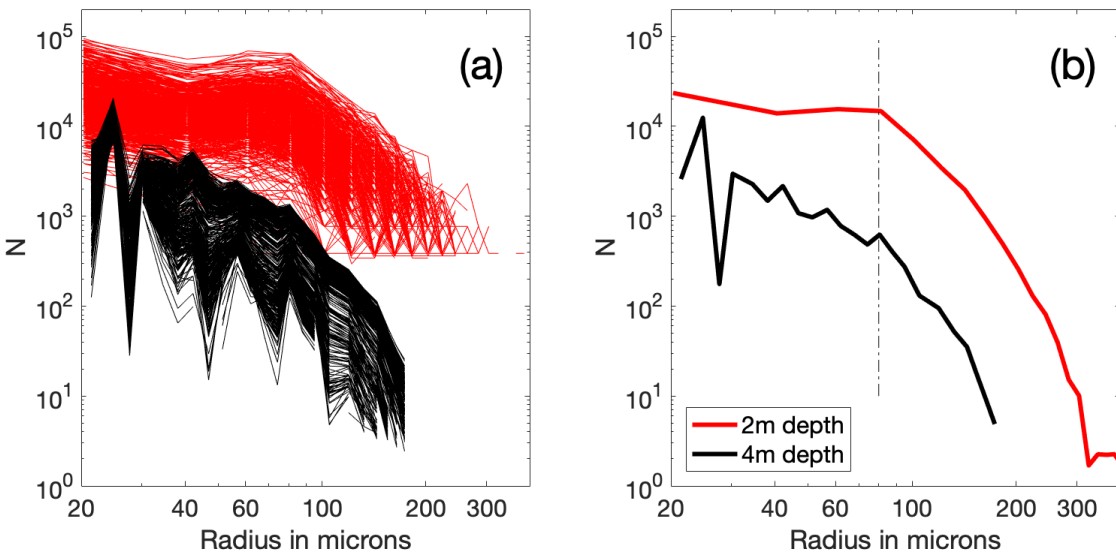

**Figure A1: The number of bubbles per unit micron radius per unit volume (N) as a function of radius. (a) all camera (red) and resonator (black) one second bubble size distributions for a single 45 minute camera measurement period. (b) the average of those distributions, including the artefact, which is the downward dip to the right of the plot. The dotted line at 80    m marks the slope break in the 2m distributions.**

.




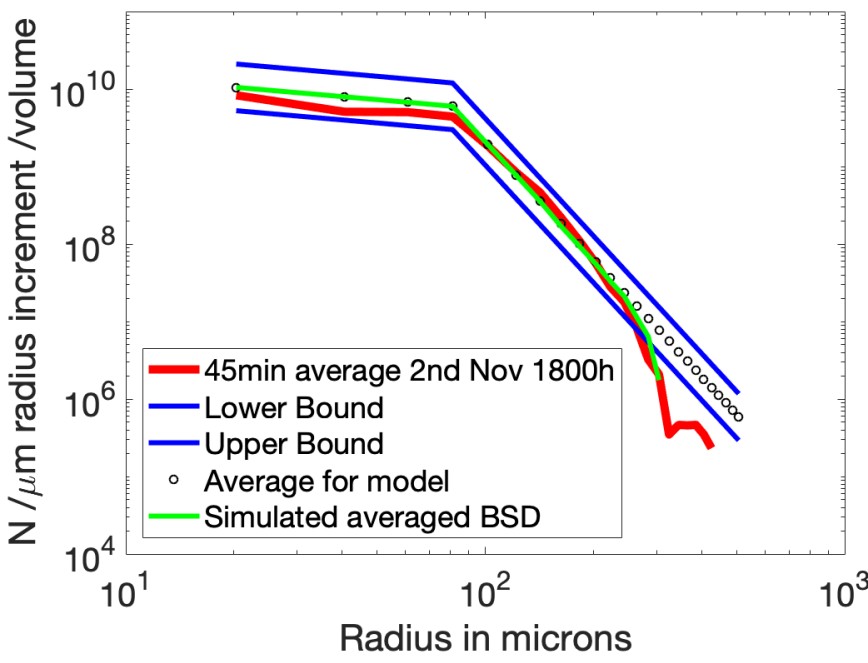


**Figure A2: A reproduction of the artefact as described in the text. The black dots are a modelled distribution with two straight slopes, and the blue lines represent a halving and doubling of that distribution (and are the same as those in figure S). This ideal distribution is scaled for each second of a real void fraction time series so that the distribution shape is maintained but the void**
**fraction matches the real data. Big bubbles that are highly unlikely to be seen over short time periods are removed as described in the text and these simulated observations are averaged (green line). The result is very close to the actual averaged data (red line).**




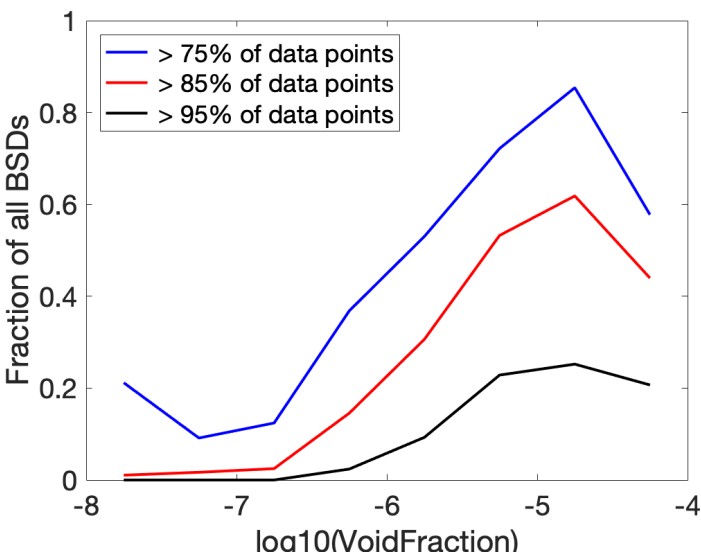

**Figure B1: The percentage of data points from each individual normalised size distribution which fall between the black lines shown in Figure S was calculated, and the proportion that pass the test for when the criteria is 75%, 85% and 95% is shown here..**

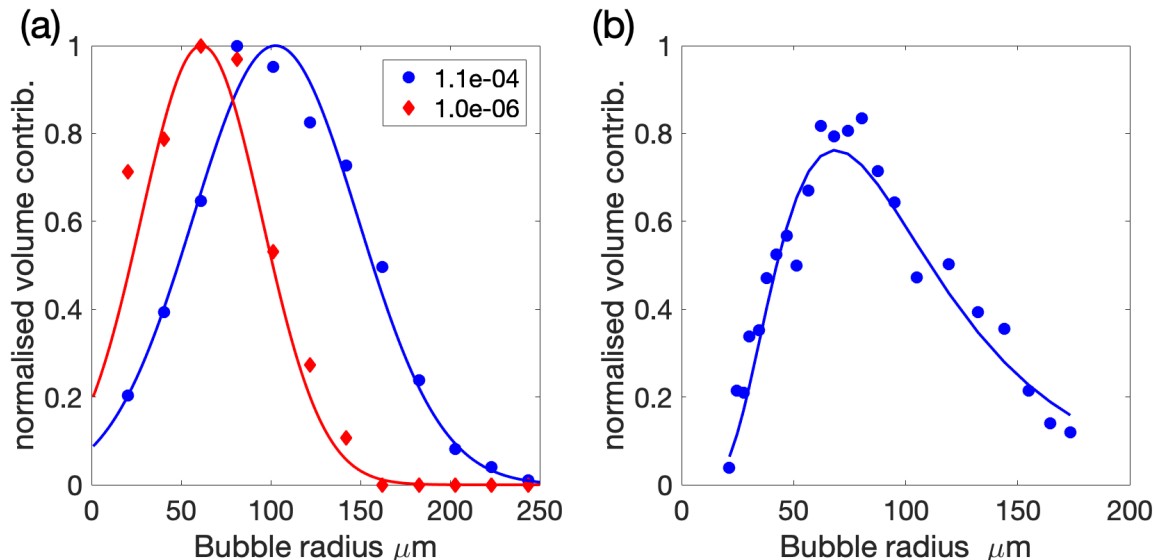

**Figure C1: Examples of fitting Gaussian distributions to normalised bubble volume distributions. The dots show the bubble numbers in each size bin and the lines are the fits. Figure (a) shows two examples (labelled by their void fraction) at 2 m depth, and (b) shows an example from 4 m with a void fraction of 4.8e-8 and a fitted peak radius of 62 microns.**




.


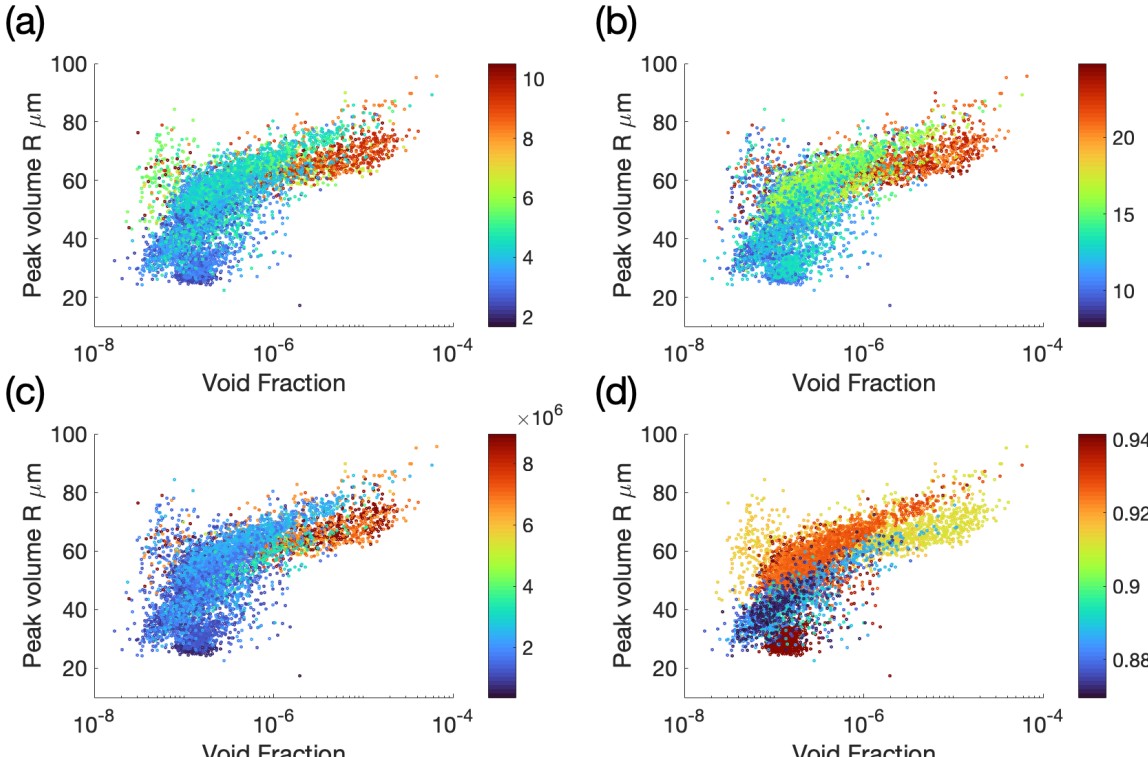

**Figure C2: Aggregations of all the data shown in Figure V, but colour coded by (a) Significant wave height Hs, (b) Wind speed (U10) , (c) Wind-wave Reynolds Number ($Re_{ww}$) and (d) estimated surface water oxygen saturation. The interpolated oxygen saturation is taken from CTD casts and so is only broadly representative of the general ocean surface on that day, and does not reflect local variations at the buoy location.**


.
