# Peer review of "Ocean bubbles under high wind conditions. Part 2: Bubble size distributions and implications for models of bubble dynamics"

_Ocean Science, 2021_

## Referee Comment (RC3)

Overall, this paper presents very interesting observations detailing bubble size distributions at depths of 2 and 4 m under high seas. The paper has significant structural problems with discussion mixed throughout the results. Also, the discussion section is rather long and somewhat repetitive. As such it needs significant revisions for clarity and other issues before the study is published.

Small item – but please use dissolved rather than destroyed, which suggests a violent and purposeful event. It seems like at line 664 you explain the concept; where you seem to propose that the bubble can maintain its size against hydrostatic pressure (like a ping-pong ball) and then collapse suddenly (as when one takes a pin-pong ball down in a ROV. Are you really implying that surfactant coatings have structural strength? This is a pretty radical proposal, and thus needs strong support.

Otherwise, the paper is very long and could really use a thorough editing to ensure that what is written is technically correct as written, that colloquial words and phrases are avoided, that run on and confusing sentences are rewritten, that duplicative material is removed, that unnecessary speculation are avoided, that discussion sentences are not in the results section (where all the data cannot be assembled to support the discussion sentences), but are in the discussion section.

This paper took me a very long time to review and as such I apologize in advance for any spelling or grammatical mistakes in the detailed items below.

– by high wind do you mean bubble formation 10 m/s or high wind as in storm > 15 m s-1. Please clarify

– Farmer et al (1993) is pretty old to use for a statement on much of the literature and was not a review….

– location is a bit vague in a moving frame of reference system (not specified).

– what do you mean by quantitative. This is a very vague term.

– Salt also strongly affects the initial size distribution or the fragmentation of the void. Not sure why you even need this (incorrect) sentence.

– you are not seriously proposing that bubble plumes in the lab are two dimensional? Even in the field, where wind and waves align, there is a directionality. Perhaps you mean boundary or wall effects.

- Some modeling studies and one citation? And what kind of models? Semi-empirical? CFD? If Fraga and Stoesser is a review, state.

– some studies have provided bubble details in this citation free declaration which really is so vague as to not actually say much of anything. Also sentence prior is a declaration without citation.

General going into problems with current knowledge before presenting current knowledge (75-94) seems backwards.

- Not sure how you know the time scale is a second. Really depends on the wave.

– Suggest you read Masuk et al 2021 10.1019/jfm.2020.933 and rephrase your comments on the Hinze scale.

– also bubble bursting.

– variability of the smaller bubbles is very vague.

– note, this is not an exhaustive list. For example, oxygen saturation or super saturation.

– bubbles also move due to turbulence

– numbers total or numbers per volumes

96-100 – words such as small and large are imprecise. Please suggest actual sizes.

– what is a partial model?

– most studies yet no citation(s).

– destruction?

– single bubble or do you mean more a group of bubbles in the statistical sense?

– platform. Was there a platform atop the buoy?

– I think there is some analysis before averaging… Also, where are the analysis algorithms described? As written Al-Lashi 2016 and 2018 are only citations for the camera.

– was the depth of the bubble camera with respect to waves known? I presume so, but it is not stated except in a statistical sense.

- $10^{-9}$ is a number $10^{-4.5}$ is a weird number. Two lines later you write $2\times10^{-7}$, why not write $10^{-6.99}$? and elsewhere. We both know the answer.

- If the noise is $10^{-8}$, then how do you measure $10^{-9}$? Are there different noise levels at different depths? Why and please report nose levels for all depths.

– comma ", which"

– 177 – you mention 2 m depth, then 4 m depth, then its unclear if you did see plumes at 2 m and none at 4 m or you saw at both.

- void faction probability distribution – maybe define this? What does the word smooth mean in this sense (not a technically defined)? what time scale are the probability distributions assessed over? I realize this is in the companion paper, but please either restrict to just the needed info for this paper (why is smooth relevant), or provide more detail.

– Figure 1 please plot (a) as semiology or loglog. Here it clearly says normalized (how and why)

figure says normalized. How was it normalized? Why was it normalized – how much does the PDF increase in reality? The normalized suggests that perhaps 0-100 μm might decrease – but I suspect not. Worst case, please put un-normalized data in supplemental. How is the PDF or histogram calculated – uniform bins?

Why is there less data at high wind speeds? Maybe a PDF of wind speeds could be useful. In any case, if you aggregate 22 and 26 m s-1 bins (Why such weird wind speed bins) you have 122 photos but a clearly decreased maximum size at 100% probability. So this explanation is possible, but not that convincing, especially given how the camera is quite shallow a non-trivial amount of time.

– 201 - Seems more of a discussion section paragraph than a results paragraph.

– its not really an artefact – which is where an analyzer fails, it is the result of time averaged size distributions being affected by a bubble-size dependent residence time. Please rephrase.

- It would be clearer to just state what the new ranges are. I also see that whereas the un-normalized ranges are similar across radii, the normalized ranges are not as consistent. Additionally, 4m, station 3 normalized clearly is two part and should be modeled accordingly. If you define a variable in the caption for the slopes, why are you not using it in the text?

– comma after break

218-219 this is repeating line 215. Delete, or rewrite. Define low bubble number (note, I don't see the lack of collapse in the data)

– true that dispersion of bubbles is important, but very small bubble change size by dilution very rapidly, so I don't see how you can rule that out. At higher densities, they dissolve slower (due to saturation) and at lower densities, small bubbles dissolve faster. Agreed that larger bubble size change is not dominant. Please revise.

– Fig. (not Fig/)

noise or variability?

252-263 – this reads more like discussion, where other arguments can be made. Suggest reducing to one sentence or just deleting here, particularly sentences that say "critical question"

delete "positions" – you already said radii

– It seems to me (by definition) due to the bend to a slope greater than -3 (S2=-3.5), this bend occurs in the uncollapsed data as well. Suggest deleting this sentence.

279-280 – agree about the replacement, not certain about the conclusion. What about uniform dissolution and higher void fraction being from more intense wave breaking not more evolved plumes? And in any case, it seems that timescales are important for this conclusion and not mentioned for this (should be in the discussion) sentence. BTW my first through is the radius of the kink has more to do with turbulence velocities and the rise velocity of a 60-80 μm bubble, with rise velocity increasing rapidly leading to low sensitivity in r with respect to turbulence. Just a thought, and in any case, the place to discuss these issues is in the discussion. Note, the smaller possible radius of the bend-over at 50 μm as in 4m station 3, is consistent with decreased turbulence motions further from the interface.

– bubble destruction? Not sure what this means as written. Do you mean loss? Dissolution? Size change is certainly not destruction.

– what does wider range of conditions mean? I guess from the rest of the para that you mean higher and lower winds? Please report what the winds are for all three data sets in the text.

301-309 –move to discussion (could varying turbulence levels explain?)

Fig. 7 weird ylabel (why not use a more standard label as in Fig. 6.

and 322 peak radius as on 318. Please be consistent.

– Suggest "far greater variability below"

– 336 – discussion. Again, could it be weaker turbulence velocities for weaker wave breaking allowing larger bubbles to escape and only keeping smaller bubbles in suspension? Move fist sentence into next paragraph.

delete "seen"

351-352 – please explain why it is clear that individual patches (two dimensional – don't you mean blobs?) each follow their own pattern? Do you mean trajectory? As written it makes no sense. Do you mean each bubble plume has a distinct pattern?

- "striking" is colloquial.

– since significant non-linearities might be expected near the surface – how was the extrapolation done? What curve fit

– how do you know the surface ocean was always undersaturated? Doesn't this depend on your extrapolation and its accuracy? And surely you don't mean even in surfacing bubble plumes (as you note in the next paragraph).

– extrapolated is not observed.

– surely you could find a citation on this matter of saturation from waves. I will see if you have one in the discussion….

Section 2.5 – limitations – this is a discussion topic.

– and found no bubbles (missing verb)

"appear to be too low for the limiting factor to be the balance between buoyancy and turbulent flow."

– accumulation of surfactants also likely plays a role. In any case, surface bubbles are advected downwards anywhere as vertical fluid motions are by definition zero at the sea-air interface (except for breaking waves).

it is not the bubble size distribution that is advected – it is the bubble plume with a specific bubble size distribution that is advected. Suggest laterally rather than sidewise which is colloquial.

– destroyed? Maybe dissolved.

– pulled downwards? Suggest advected downwards

– delete gradually – you don't know if it is or is not gradual, and in any case, what is gradual? Not defined.

– "cannot last long enough" specify, please. Dissolve and/or rise too rapidly to form a background population.

– what do you mean by heterogeneity of the bubbles? Normal meaning is spatial heterogeneity; however, this makes no sense in this sentence.

487-494 this feels like a repetition of text earlier in this section. In fact, I think this whole section can be re-organized and shortened and better focused.

– note, when bubbles burst at the sea surface it creates new microbubbles.

– can you provide some guidance on what is a large bubble (r>??)

– you are not really suggesting anything. Perhaps better to explain this as a knowledge gap – proposing an unknown process or processes or series of processes but not proposing what they are is not actually proposing something.

– mix – I think you mean diffuse….

516-518 – citation needed (e.g., the work of Bruce Johnson)

– probably also the type and concentration of surfactant – not just its presence or absence – particularly given how much surface area is created by a bubble plume.

– destroyed is what a military does to a target with a missile! And if you check out the Johnson paper, you do not know if the bubble really is destroyed or shrinks to a stabilized microbubble of a few microns in a surfactant matrix. Just say dissolve or burst at the sea surface.

525-527 – these two sentences are repetitive and could be combined and shortened.

– why would there be a thin bubble free layer below the sea surface? This seems unlikely given that buoyancy does not disappear. Since it is speculative with no data and seemingly non-physical, suggest delete.

– suggest advect not pull.

– only bubbles with rise velocities less than the downwards velocity.

– what is the evidence that you larger bubbles are not trapped in the centre than edges. If this was in the companion paper it needs to be cited. Also how much larger is larger? This is a very qualitative assessment.

– destruction? I presume you mean dissolution.

also could be (split verbal construct)

– what about stabilized microbubbles? More accurate to say the vast majority or almost all, or something not so definitive. Also write in present tense, not passive past tense. Microbubbles could diffuse to the sea surface from those depths. Plus Langmuir circulation is not just down…

– forms

– all these bubble are injected into the ocean? They are already injected into the ocean!

556-569 – very long and convoluted sentence.

– as written this is not correct. The force on the bubbles is always in equilibrium. Furthermore, its hard to see how bubble dissolution could decrease saturation state. A very clumsy sentence grammatically. In any case, I am not certain if this section is really needed – it doesn't make many points and the one that it does – mixing lowers void fraction – which could be merged into other discussion sections.

-or they are produced rapidly and dispersed widely or transit to deeper depths and thus have limited residence time at this depth – Farmer ad Li (1995) show bubbles from Langmuir circulations to 10 m.

– we suggest that bubbles collapse . . .  This is speculative, so please propose a new mechanism as dissolution is quite well understood and is not consistent with a sudden collapse. Or delete. Additionally, any talk of sudden and slow needs to specify what timescales are slow or sudden.

600-605 – The increase of hydrostatic pressure with depth (and decreasing saturation) is not new and is not a process that would be termed sudden collapse.

- Open questions should not be after conclusion! This seems more like a future research section in the discussion. Also, all these points have been brought up throughout the discussion section and even the paper, so this is a bit of a repetition – I think it is good to summarize them all somewhere, but then lessen the number of times these are mentioned throughout the paper.

i.      This is a long open question – maybe break into two. You have measurements of two environmental conditions – introduce with your findings and then be more specific on your open question ii.     Couldn't you answer this with sonar data? At least roughly?

iii. timescales of advection at convergence zones. – you report 20-40 s, so this is not an open question or it is poorly phrased.

vi. nature of bubble coating – this was really not addressed in the paper, so seems beyond your paper's scope.

Viii (where did vii go?)

---

## Author Comment (AC1)

We are grateful to the reviewer for their comments and have prepared a revised manuscript which addresses their concerns. Our response to each point and a description of the changes made are given below (reviewer comment in blue, author response in black).

The paper presents results from the field campaign highwings reporting on the bubble measurements in the first 10meters below the sea surface using optical and acoustic techniques. The paper is important for the community as very few of such measurements exist. I have a set of recommendations or suggestions for the authors to improve the manuscript.

In particular, I strongly recommend to use the published data from highwings on gas transfer, waves and whitecap coverage and see the correlation with the bubble plume. The data are available online and have been used by various authors . The bubble plume measurements are by construction fragmentary since the full entrainment process is not captured but it is certainly a very valuable/important information. Combining this information together with the gas transfer velocity, the whitecap coverage and comparing with recent models and parameterization would shed light on the transport process of the bubbles and how the present data can be used to better constrain these models.

We understand the desire to see an analysis of bubble data in relation to air-sea gas transfer data. We are well-aware that the data exist – two of the authors of this paper are also co-authors on the HiWINGS gas transfer publications. We do not present that analysis here for these reasons:

- Firstly, we note that we do make comparisons with wave parameters, both in the companion paper (referenced multiple times in this paper) and in Appendix C of this paper, where we show that the bubble data does not show clear patterns when segregated by the wind-wave Reynolds number or the significant wave height.
- The bubbles produced by breaking waves are important for several different fields: aerosol production, optics and underwater acoustics as well as air-sea gas transfer. Our aim here is to present basic data on bubble size distributions which is useful to all the fields that need it, rather than bias the presentation by tuning it to a particular question. There is a severe lack of basic information about the composition of deep bubble plumes, and addressing this is the priority.
- A follow-on study could indeed compare the air-sea gas transfer and the bubble parameters. However, this is a significant separate piece of work, and this paper (which, as noted above, covers extensive details about bubble plumes which have not previously been observed) is already close to 12,000 words long. We do not consider that the community would be well-served by us removing details of the basic data analysis in order to make room for gas transfer comparisons. In short, a gas transfer

paper would be desirable but would be presented in a different paper.  We also note that it would be based on the fundamental observations presented here, which necessarily come first.  We are seeking the resources to do that additional study, although since all the data is public others could also take on that task.

- Our bubble data was measured at depths of 2m and below, and it is generally accepted that the majority of $CO_2$ uptake occurs very close to the surface. As we state in the paper, we think that a full understanding requires measurement of the flow processes and the dissolved gas spatial distribution in addition to the bubbles, and we think that a simplistic comparison that ignores those complications is unhelpful.
- We also note that there are extensive assumptions made in modelling studies about the structure, composition and movement of bubble plumes, but very little available data to base those assumptions on.  We therefore consider that a first step towards helping the modelling community is in providing constraints from field data, which can be used to modify and validate their models.

.

In the introduction, you state "If normalised by void fraction, these distributions collapse to a very narrow range, implying that the bubble population is relatively stable and the void fraction is determined by bubbles spreading out in space rather than changing their size over time." Back of the envelope calculations of bubble mediated gas exchange can tell you that the transfer of $CO_2$, $O_2$ is relatively slow compared to your observation time of the bubbles, so that bubbles will not change size significantly in the first few meters even if they are exchanging gas (very true for $CO_2$ which does not account for much of the volume). Bubbles will only change size if they are brought deeper in the flow by some turbulence process (Langmuir turbulence, etc). This is discussed in the earlier papers from Keeling 1993, Woolf and Thorpe and is implicit in Liang et al 2011, 2012  or Woolf' modeling. I would recommend looking at recent modeling work on that topic:

Leighton TG, Coles DG, Srokosz M, White PR, Woolf DK. 2018. Asymmetric transfer of $CO_2$ across a broken sea surface. *Sci. Rep.* 8:8301

Liang J-H, McWilliams JC, Sullivan PP, Baschek B. 2011. Modeling bubbles and dissolved gases in the ocean. *J. Geophys. Res.* 116:C03015

Liang J-H, McWilliams JC, Sullivan PP, Baschek B. 2012. Large eddy simulation of the bubbly ocean: new insights on subsurface bubble distribution and bubble-mediated gas transfer. *J. Geophys. Res.* 117:C04002.

We are aware of no direct in-situ measurements of bubbles changing radius through dissolution in the top few metres of the ocean, and only one attempt (Czerski, 2011) to directly characterise the bubble coating in the open ocean.  The parameters at play are understood - bubble radius, surfactant coatings, the possible contribution of particulates to stabilisation, pressure and the local gas saturation – but there is no actual data from the ocean to tell us whether the bubble models currently used balance these influences in the right way. Our data set is valuable precisely because it includes the most detailed measurements of bubble size distributions in high winds to date, and this can be used to test models. Our approach here is to avoid assumptions, and to base our analysis on field data (ours and that of other people) only.  So we think it important to make statements like the one the reviewer is commenting on because this is what actual measurements imply, unclouded by the

assumptions often made for the purposes of modeling.  If our data support the models, the point is not that we are making comments that lack novelty, but that when faced with actual field data, the models match the measured reality.

We have looked at the three papers suggested by the reviewer.

The Leighton 2018 paper discusses models which are based on minimal data, with significant limitations in the data collection technique that are not addressed or discussed (either in the main text or the supplementary material). The data that we present from our study is significantly more robust, with methods and the limitations from environmental conditions carefully described.  We do not see any reason to re-interpret our results based on this 2018 study, when it is far more limited in scope and quality than our own.

The Liang 2011 and 2012 papers develop a bubble model which presents a similar overall picture to our data, and we are grateful to the reviewer for bringing these to our attention. However, we note that they are not directly tested against field data except in a general way, and there are places where the model outputs do not agree with our data (which is perhaps not unexpected considering the breadth of the model and the limited field data available when they were written).  We have added these papers to our literature review, and we have also added a brief comment on the comparison between their data and ours at line 558.

Towards the end of the intro, you state: "We suggest that as bubbles move to depths greater than 2 m, sudden collapse may be more significant as a bubble destruction mechanism than slow dissolution, especially in regions of high void fraction."  Yes, this is discussed by modeling studies from Liang et al and Woolf et al. The Langmuir type entrainment process is necessary to bring small bubbles down where they can collapse due to hydrostatic pressure. However, the life of the larger bubbles in the top two meter is important for co2 transfer, they exchange gas during their lifetime without changing the bubble size, since the content of $CO_2$ is small compared to the overall volume of the bubble. This is discussed or implied in the models by Keeling 1993 and then Deike and Melville 2018

 As we emphasize above, our aim here is to present experimental data by itself without relying on the interpretations that have come from modelling.  We think that it is important to be clear about the implications of our data, and we note that one of the other referees thought that collapse was a novel concept that required strong additional justification, suggesting that this possibility is far from accepted.  We are not making claims to novelty here; we are stating the conclusions from our own field data.  We also note that the word "collapse" does not, as far as we are aware, appear in any of the papers the reviewer mentioned, except when used to describe the collapse of data sets on to a single line, or for the deformation of large air pockets during the breaking process.  It may be implied and obvious to those creating the models, but not apparent to others. We are talking about a sudden process happening at depth, and we were not aware of this suggestion being made explicitly elsewhere in the literature.  We want the language we use here to be clear and unambiguous.

I would recommend citing Bowyer PA. 2001. Video measurements of near-surface bubble spectra. *J. Geophys. Res. Oceans* 106(C7):14179– 90; which present of the few direct optical measurement of bubbles below the ocean surface.

We have added this citation on line 80.

I do not understand the statement: "Our results suggest that local gas supersaturation around the bubble plume may have a strong influence on bubble lifetime, but significantly, the deep plumes themselves cannot be responsible for this supersaturation"

In conversation with other researchers in this field, we have noticed a strong tendency to assume that the deep plumes are significant because the gas contained in those bubbles will be injected into the ocean at depth and will significantly increase the gas saturation in the water at that depth.  Our point here is that it seems unlikely that there is enough gas in the bubbles in the deep plumes to make a significant difference, but we think that the bubbles may continue to exist because they are in a water packet with higher than average gas saturation.  We are suggesting that the main gas contribution associated with a deep plume is already in the dissolved phase, rather than the gaseous phase.  This is explained further in the main text at line 425.   We have changed the phrasing in the abstract to clarify our meaning and it now reads "… *significantly, the gas in the bubbles contained in the deep plumes themselves cannot be responsible for…*". (line 30).

The authors present data on bubble void fraction, distribution, etc. During the same campaign, the gas transfer velocity (or piston velocity) for CO2 and DMS has been measured and is reported in Brumer et al 2017 (GRL) (which is not cited), as well as whitecap coverage data. Could the authors present cross comparison of these quantities? Similarly, Deike et al 2017, and then Deike and Melville 2018 presented scaling for air entrainment by breaking wave and some of that has been used to predict gas exchange. While your data do not get the full air entrainment because you are missing the large bubbles close to the surface it would be interesting to see whether the proposed scaling in the literature in terms of dependency with wind and wave can work or not. Similarly, Brumer et al 2017 proposed a wave Reynolds number scaling for gas transfer and it would be interesting to see if your bubble data follow such scaling. Finally seeing the correlation between whitecap coverage data from Brumer et al 2017 and your bubble data would provide information on how much of the bubbles are being transported down to the depth where you are making the measurements. This could be very useful for future modeling on the role of small bubbles getting fully dissolved in the water column, which will contribute to exchange of less soluble gases such as O2, N2.

Brumer S, Zappa C, Blomquist B, Fairall C, Cifuentes-Lorenzen A, et al. 2017a. Wave-related Reynolds num- ber parameterizations of CO2 and DMS transfer velocities. *Geophys. Res. Lett.* 44(19):9865–75

Brumer SE, Zappa CJ, Brooks IM, Tamura H, Brown SM, et al. 2017b. Whitecap coverage dependence on wind and wave statistics as observed during SO GasEx and HiWinGS. *J. Phys. Oceanogr.* 47(9):2211–35

Deike L, Lenain L, Melville WK. 2017. Air entrainment by breaking waves. *Geophys. Res. Lett.* 44(8):3779– 87

Deike L, Melville WK. 2018. Gas transfer by breaking waves. *Geophys. Res. Lett.* 45(19):10–482

We are well aware of the gas transfer data from this cruise, as stated above in our response to the general points.  We have provided our reasons for not making extensive comparisons to the gas transfer data there: that is a separate piece of work.  This is a more fundamental paper dealing with the bubble measurements, and any gas transfer comparison would build on the work presented here.

About figure 2: the shapes at 2m seem compatible with the modeling work from Liang et al; which starts from a Deane and Stokes like distribution - similar to other recent bubble gas transfer formulation. While a quantitative comparison is probably out of scope, this should be mentioned. It seems that the present measurements are compatible with the current assumptions from bubble models.

We assume that the reviewer is referring to figure 6 in Liang 2012, since this is the only plot we can find that refers to a depth of 2m, although the wind speed for those models is only 10 m/s.  According to the data in our companion paper, the dominant void fraction at 10 m/s wind speed and a depth of 2m is $10^{-7}$, so the relevant comparison is the bubble size distribution shown in blue and turquoise lines in figure 4(a) of this paper.  We have taken the $10^{-7.5} – 10^{-7}$ void fraction size distribution for this wind speed range, multiplied each point by the radius cubed, and to the left we show the overlay of the two plots.  The comparison is between the 2.15 m dashed line and the blue one.

[Figure]

The discrepancy here is about a factor of 100, although the shape is similar.  This is clearly a relatively crude comparison, but it seems a stretch to say that the model is compatible with the data.  We have not added a comment to the paper, since this comparison is limited but does not suggest that the model outcomes are appropriate in this case.

Similarly and about all figures on bubble size distributions. Can you plot your integrated volume from these distribution as a function of wind speed? void fraction? wave age? wave height? Whitecap coverage? Gas transfer velocity? This could be useful to compare with existing models from air entrainment based on breaking dynamics (assuming the air lost

between entrainment and 2m down scale in a similar way which is not obvious at all). This would provide very useful information/constraints for modeling.

The companion paper (now accepted for this journal, https://doi.org/10.5194/os-2021-103) focusses on void fraction distributions and how they change with wind speed, wave age etc. This data is all presented there (with the exception of gas transfer comparisons, for the reasons stated above).

---

## Author Comment (AC2)

We are grateful to the reviewer for their comments and have prepared a revised manuscript which addresses their concerns. Our response to each point and a description of the changes made are given below (reviewer comment in blue, author response in black).

The manuscript reports on a large data set of bubble size distributions at 2m and 4m depth, and in various wind and wave conditions. The data set is extensive and is undoubtedly of interest to the community. The paper merits being published after revisions.

General comments:

I appreciate the difficulty of making such measurements and the dataset is certainly impressive. While I appreciate the honesty of the authors in clearly stating the limitation of the data set, I find the analysis somewhat speculative and lacking, particularly in the use of the ancillary data available from these measurements campaigns.

We are not sure which ancillary data the reviewer is referring to here, but much of it (ADV data, sonar and wave measurements) is discussed in the companion paper. We have added text on line 135 to clarify this point.

It would be interesting to see void fraction and the peaks of the moments of the bubble size distribution, as a function of wind speed, and wave related parameters such as significant wave height, wave age, significant wave slope, and breaking probability (white cap coverage and the like). The data is sufficiently wide that decencies on the sea state are likely to emerge.

The companion paper (which is referenced throughout the text where its results are relevant) is entirely focussed on void fraction measurements and provides a full analysis of that data with respect to wind speed and wave-related parameters. This is mentioned at the end of the Introduction, and in the discussion of the results. We have added text at line 524 to reinforce the point that the discussion and conclusions are based on the data from both papers, and that we are making scientific judgements based on both sets of results. We also note that Appendix C already shows segregation of the data by significant wave height, wind speed, the wind-wave Reynolds number and oxygen saturation, and no notable relationships are seen. We think that the major revisions the reviewer is requesting are all covered in the companion paper and so we have made no amendments to this paper in response to this comment other than reinforcing the references to the other paper.

Minor comments:

Line 68, you mentioned that current numerical models cannot yet reproduce the complexity of breaking and air-entrainment events. I think they are getting pretty close and that's worth mentioning. See Deike et al, JFM 2016 for example. Although it is acknowledged that these numerical models do suffer from the same scale limitations as laboratory experiments do.

We acknowledge the progress represented in the Deike 2016 paper, but that paper is focussed on the total entrained volume of air rather than the details of the bubble processes after entrainment. We have added references to more modelling studies (Deike 2016, Liang 2017) and have made the current progress and limitations clearer. We have also added a note about the lack of field measurements of subsurface flow fields and gas and bubble spatial distributions to validate the assumptions used in those models.

If understand correctly, figure 1a shows the probability of measuring a bubble of a given radius for different wind speeds; and figure 1c shows the radii in the very tail of these distributions in the 90$^{th}$ percentile and up. Does this mean that figure 1c shows bubble sizes that are essentially unlikely to be present in the data (90% of the time)?

This is correct. We have displayed the data in this way because different readers infer different meanings from "maximum bubble radius" – there are always going to be outliers, and we think that presenting the data at a range of percentiles shows where the meaningful cut-off is and allows readers to draw their own conclusions. We think that the 99$^{th}$ percentile is perhaps the most useful for comparison with the turbulence data, ignoring the most extreme outliers but incorporating the bulk of the data. The question here is "what is the largest bubble size that the physical processes allow to be present at this depth?", and so the extreme tail of the distribution is relevant. We have added text to clarify that Figure 1(c) shows data relevant to the tail of the distributions in Figure 1(a).

For clarity, please use the same labels for the normalized size distributions of figures 2b, 3, and 4. Also, it is important to note that the normalization does not render the data dimensionless.

We have re-made the figures with consistent y-axis labels, including the dimensions for the normalised data. We have also modified the captions to match.

Please, rephrase line 275 "At 2 m, the peak volume has a limited relationship with the void fraction, and does not show a large decrease immediately after a peak." I believe the authors meant to say something akin to "At 2 m, the radius of peak volume has a weak relationship with the void fraction, and does not show a large decrease immediately after a large void fraction events"

We have amended the text as suggested.

---

## Author Comment (AC3)

**Response to Referee #3**
**"Ocean bubbles under high wind conditions. Part 2: Bubble size distributions and implications for models of bubble dynamics" by Helen Czerski et al., Ocean Sci. Discuss.**
**https://os.copernicus.org/preprints/os-2021-104, 2021**
**March 2022**

We are grateful to the reviewer for their detailed comments. Our response to each point and a description of the changes made are given below (reviewer comment in blue, author response in black). We have prepared a revised manuscript which addresses their concerns.

Overall, this paper presents very interesting observations detailing bubble size distributions at depths of 2 and 4 m under high seas. The paper has significant structural problems with discussion mixed throughout the results. Also, the discussion section is rather long and somewhat repetitive. As such it needs significant revisions for clarity and other issues before the study is published.

In presenting these results, it was necessary to cover several different types of analysis for a very large data set. We made a deliberate decision about the structure: to present an immediate analysis with each set of results in order to explain their significance and to set up the next tranche of results, and then to use the discussion and conclusions to explore the meaning of all the results together. Our judgement was that leaving all of the discussion until the end would make this long paper much harder to read, since reading the primary discussion for each figure would entail going back and forth to figures many pages back. We have checked the discussion for repetition and eliminated it where possible. We have also added text at the start of section 3.3 to emphasize a point that was not sufficiently clear in the first version – that the conclusions at the end are based on both the results of this paper and also the companion paper (now accepted) which covers void fraction analysis of the same data set.

Small item – but please use dissolved rather than destroyed, which suggests a violent and purposeful event. It seems like at line 664 you explain the concept; where you seem to propose that the bubble can maintain its size against hydrostatic pressure (like a ping-pong ball) and then collapse suddenly (as when one takes a pin-pong ball down in a ROV. Are you really implying that surfactant coatings have structural strength? This is a pretty radical proposal, and thus needs strong support.

Our aim here is to present our experimental results in a way that reflects previous at-sea observations but which does not make assumptions. The ocean bubble literature almost universally makes the assumption that bubbles always grow and shrink slowly, but we are unaware of any direct measurements of this at sea. Johnson and Wangersky (1987) showed that bubbles that are coated with both surfactants and particulates can maintain their size for a significant period and then collapse suddenly when the pressure increases. To our knowledge, there are no in-situ measurements of the coatings surrounding ocean bubbles, with the exception of one paper that shows that the coatings may be much stiffer than previously expected (Czerski, 2011), even in highly oligotrophic waters. There is therefore no knowledge about whether the gels and particulates present in the water are also stabilising bubbles, and no reason to assume that surfactants are the only substances that adhere to bubbles.  Studies in other areas (for example food science) suggest that even small levels of particulates have a highly stabilising effect. If rapid collapse does occur in some situations, we are not suggesting that the prior stabilisation is due to surfactants alone, but to surfactants combined with particulates.  This is our conclusion based on the evidence we have, which we arrived at without starting from the generally accepted, but unsupported, assumption.  We accept that this is a significant departure from the previous assumption, but our evidence suggests that it merits serious consideration in some circumstances.

Our data provide some support for slow dissolution processes at low void fractions (figures 6 and 7), but this does not explain the almost identical radius at the volume peak at both 2m and 4m.  If slow dissolution was at work, we would expect to see wide variation in the bubble size distribution for plumes of different ages, whereas what we observe is the fixed bubble size distribution shown in figure 4(a). This suggests that the bubbles are reasonably stable over their lifetime in the plume. We have changed the wording to "terminate" rather than "destruction", but we are indeed suggesting that as bubbles descend in the water column, their transition from a relatively stable size to being completely dissolved may be a very rapid process. We have added some sentences at line 664-666 to clarify our position on this point.

Otherwise, the paper is very long and could really use a thorough editing to ensure that what is written is technically correct as written, that colloquial words and phrases are avoided, that run on and confusing sentences are rewritten, that duplicative material is removed, that unnecessary speculation are avoided, that discussion sentences are not in the results section (where all the data cannot be assembled to support the discussion sentences), but are in the discussion section.

The paper is long out of necessity; we are presenting detailed results which will be of interest to researchers in a wide range of disciplines because measurement opportunities like this are rare. This will be the most detailed in situ ocean bubble data in the literature to date, and we think that it justifies taking the space to present a full analysis.  We have tried to make the text as concise as possible while fulfilling those aims, and we have continued to edit with that in mind while dealing with reviewer responses.  Our responses to the more specific points are below, but we believe we have justified our original text in many cases.

This paper took me a very long time to review and as such I apologize in advance for any spelling or grammatical mistakes in the detailed items below. I could have accomplished the review quicker by just generalizing issues to address, but am very supportive of the science being published and thus spent the time.

– by high wind do you mean bubble formation 10 m/s or high wind as in storm > 15 m s- Please clarify

We have changed this phrase to "by breaking waves".

– Farmer et al (1993) is pretty old to use for a statement on much of the literature and was not a review….

We included four papers here which deliberately covered 30 years of work, to support the statement *"… have been studied for many years"*. Google Scholar suggests that the Medwin & Breitz 1989 paper has been cited 159 times, the Farmer 1993 paper 225 times, the Graham 2004 review 25 times and the Vagle 2010 paper 65 times. We consider that together they are reasonably representative of the progress made in this field over the past 30 years, with the early papers representing significant advances, and so we have not made any changes.

– location is a bit vague in a moving frame of reference system (not specified).

This is a general introduction to the paper, and "location" is intended to refer to instantaneous position on the spatial path that bubbles take as they progress from creation to destruction, which is not known in any frame of reference. We understand that there are subtleties with respect to moving frames of reference, but elaborating that point here would break the flow of the argument at this point, which is focussed on gaps in knowledge. We have changed the phrasing to "*location within the water column*".

– what do you mean by quantitative. This is a very vague term.

We use "quantitative" here to make the distinction from "qualitative" data. Much of the previous data on bubble plumes has been at least partially descriptive (and therefore qualitative), because it is challenging to characterise complex 3D structures, because of the variety of measurement techniques used and also because the lack of a standardised way to characterise bubble plume shapes. It is also partly due to a lack of clarity about which features are most important to measure, which can only come from better mechanistic understanding of the processes in action (and which is the aim of this paper). We have therefore left this phrase as it is, because we consider that it's the most concise way to describe the need to clarify the essential features of ocean bubble formation and distribution, and then provide quantified measures of those features at sea.

– Salt also strongly affects the initial size distribution or the fragmentation of the void. Not sure why you even need this (incorrect) sentence.

We are aware of no unambiguous evidence in the literature that salt changes the outcome of the turbulent fragmentation processes that are responsible for creating the initial bubble size distribution at sea. However, evidence that salt changes the way the bubble plumes evolve *after* formation can be found in the literature.

Slauenwhite & Johnson did see differences in bubbles generated at an orifice in fresh and salt water, but the relevance of this method of bubble production to processes at sea has been questioned by Garrett (2000). However, papers specifically studying the processes of fragmentation in turbulence (Deane & Czerski 2008, Czerski & Deane 2010) compare the details in fresh and salt water and find the outcomes to be almost identical in all cases. Loewen (1996) and Callaghan (2014, figure 7) show that the initial air entrainment in a wave tank in freshwater and artificial seawater (salts only, no organic material) is almost identical. We mention salt here because it is well established that salt does prevent bubble coalescence (as described in the papers referenced), and therefore it is likely to have an influence as the bubble plume evolves. Our point here is that wave tank measurements conducted in salt water are more relevant to the ocean than those conducted in freshwater, and that the differences are seen as the plume evolves but not as it is initially created.

– you are not seriously proposing that bubble plumes in the lab are two dimensional? Even in the field, where wind and waves align, there is a directionality. Perhaps you mean boundary or wall effects.

Our intention was to highlight the overall geometry of the wave, and this phrasing is commonly used in the literature (for example Callaghan 2014, Deane 2016 and Rojas 2010). In a wave tank, the breaking process is designed so that (apart from small edge effects) the waves have an identical profile at every position across the tank until the point of breaking. This is not representative of the open ocean, where the geometry is much more complex. There are now tanks designed to create more realistic waves (for example the multi-directional wave basin at Imperial College in London), but the vast majority of existing studies have been conducted in tanks which created a 2D wave profile, not a 3D one. We believe that readers of this paper will be well aware of the intended meaning here, and that the current phrasing is the most concise way of making this point. We have not changed the phrasing here, since this usage is consistent with the broader literature.

- Some modeling studies and one citation? And what kind of models? Semi-empirical? CFD? If Fraga and Stoesser is a review, state.

The Fraga & Stoesser paper is not a review. It is a study using a large-eddy simulation based Eulerian-Lagrangian model to examine bubbly plumes in general (including sudden gas release in a variety of situations), and we chose it as an example of the modelling efforts on multiphase flows in general, since there are relatively few on the specifics of the plumes caused by breaking waves. Our paper is not focussed on modelling and so we had chosen to limit the number of articles cited in order to keep our paper concise, but we have added three more relevant papers as references here.

– some studies have provided bubble details in this citation free declaration which really is so vague as to not actually say much of anything. Also sentence prior is a declaration without
citation.
General going into problems with current knowledge before presenting current knowledge (75-94) seems backwards.

We assume that the sentence referenced here as line 71 is this one: "*The combination of open ocean and laboratory experiments has produced a general overview of the generation and development of bubble plumes immediately following on from breaking waves, but details at the scale of individual bubbles are lacking.*". The first clause is a summary of the paragraphs that immediately precede it, and so the citations are the ones presented in those preceding paragraphs.

We have modified the end of that sentence so that it now reads like this: "*but a full mechanistic understanding requires details of the processes influencing individual bubbles.*", to prevent pre-empting the outcome of lines 75-94.

- Not sure how you know the time scale is a second. Really depends on the wave.

Deane and Stokes (2002), referenced in this sentence, point out that it is possible to tell when bubble fragmentation ceases because the bubbles become acoustically quiescent (figure 1 in that paper & the preceding text). Figure 3 in that paper shows the acoustically active period of bubble formation lasting for one second, and they say "*One of the problems with existing measurements of oceanic bubble size distributions is that the bubble spectrum evolves quite rapidly in the first second or so after the active phase of air entrainment ceases*". Subsequent papers (eg Callaghan 2013) also show the active phase lasting for one second or less. We take the reviewer's point that at sea the spilling process can continue for a few seconds as progressive parts of the crest overturn, and so we have changed the phasing to match that of Deane & Stokes: "the first second or so". However, at any single location, the evidence suggests that the active phase does not last beyond one second.

– Suggest you read Masuk et al 2021 10.1019/jfm.2020.933 and rephrase your comments on the Hinze scale.

We are grateful to the reviewer for drawing our attention to the Masuk paper. Another very recent paper (Riviere 2021) has addressed a similar issue from a theoretical point of view. We have added both references to our paper, but only include a brief comment since these large bubbles are not the focus of our paper.

– also bubble bursting.

We assume that the reviewer is referring to the sentence that starts "*Above this size, turbulence can cause bubble fragmentation...*". We are confused by this comment, because "bubble bursting" in this field refers specifically to the situation where bubbles have risen to the surface so that they develop a curved film or cap which protrudes from the surface into the air. Bubbles are said to burst when this curved film either breaks or is punctured (see for example Deane 2013), and this process has been studied in the case of ocean bubbles because it may produce aerosols (De Leeuw 2011). It is clear in the paragraph containing line 79 that we are referring to sub-surface turbulence, and fully submerged bubbles cannot burst – bursting is purely a surface phenomenon. We have never seen any reference to the idea that subsurface turbulence could have a significant effect on bubbles bursting at the water surface. We have made no changes in response to this comment.

– variability of the smaller bubbles is very vague.

We assume that the reviewer is referring to line 86, which says "*There are still many open questions associated with this initial period of bubble formation, particularly the variability of the smaller bubbles…*".   We do not want to extend a paper which is already (as the reviewer has noted) very long, so we have added a reference to a 2022 review paper (Deike 2022) which includes a figure showing the variability in the existing sub-Hinze scale measurements, and changed the phrasing to "*particularly the variability of the size distribution of smaller bubbles…*".

– note, this is not an exhaustive list. For example, oxygen saturation or super saturation.

This sentence refers to "*this initial period of bubble formation*", and as described in the text, this refers to the initial entrapment of air pockets to form bubbles, bubble fragmentation over a timescale of a second or so in response to temporary intense turbulence, and the formation of tiny bubbles due to splashes, jet impingement and Messler entrainment.  All of these processes take place over a similar timescale of a second or so, as described in response to the query about line 74.  These are fluid flow processes that create air pockets of different sizes very rapidly to form the initial bubble population, and we are unaware of any sources in the literature that suggest that gas saturation level makes a significant difference to these processes on these very short timescales.  We have not made any changes to the text here.

– bubbles also move due to turbulence

The sentence here reads "*Once formed, bubbles move due to buoyancy and advection..*".  The definition of advection is *transport [of an object] by the movement of a mass of fluid.* Bubbles being moved around by turbulent flow is therefore included in the definition of advection, in addition to advection caused by larger scale organised flows.  We also note that many other authors (Deane 2016, Goddijn-Murphy 2016, Liang 2012 among many others) have used this term in exactly this way, for example:

"*The model successfully simulates essential bubble processes including gas dissolution, size change, buoyant rising and advection by turbulent flows.*" – Liang, 2012.

We have made no change to the text here.

– numbers total or numbers per volumes

The text here reads "*They also found some evidence that numbers of large bubbles (>200 $\mu$m in radius) at a depth of 0.5 m…*".  The abstract of the paper cited includes this sentence: "*Also, the number of bubbles with radii >200 mm is found to be dependent on the heat flux*", so our phrasing here directly reflects the phrasing of the cited paper's authors.  They are referring to daily averaged bubble size distributions (figure 10 in their paper), and since there is no way to know whether this is dominated by a small number of long-lasting bubbles or a large number of short-lived bubbles, it is not possible to comment on whether it is the total number of bubbles. We consider that it is best to leave the phrasing as it is, because this reflects the peer-reviewed statement made in the paper we cite.

96-100 – words such as small and large are imprecise. Please suggest actual sizes.

We refer here to the two measured ends of the bubble size distribution, which are always in similar places: around 2-4 mm radius for the "large end" and below 100 μm for the small end. We did not include specific numbers because what we are talking about is the general principle of what happens at the large and small extremes of a bubble size distribution. There are no specific boundaries: the larger a bubble is, the more likely it is to rise all the way to the surface, and the smaller a bubble is, the faster it is thought to dissolve. We consider "large" and "small" to adequately reflect this. We have made no change here.

– what is a partial model?

The text here reads "*Deane et al constructed a partial model…*".

The conclusions of Deane's paper include this text: "*The model calculations presented here are incomplete in the sense that the spatial and temporal evolution of wave-induced turbulence from individual wave breaking events has been treated only indirectly. … A more complete physical description of the problem would include a model for the turbulence decay rate and the wind-speed dependence of wave breaking rate together with the statistics of breaking wave kinematics and a specification of vertical water motions associated with buoyancy flux and Langmuir circulation.*".

We consider that "partial" is an accurate and concise representation of this situation, particularly since we describe the limitations in the next sentence: "… *This model was designed for the evaluation of the acoustics of the bubbly water near the surface and did not contain an explicit bubble source function, but matched observations of acoustical attenuation at sea*".

We have changed the text here to read "…*constructed a model that partially described the properties of the larger bubbles…*". We have also extended this description of the limitations to read "*… and did not contain an explicit bubble source function or a complete description of near-surface flow patterns and wave breaking..*"

– most studies yet no citation(s).

We have added citations to back up this specific point: Trevorrow 2003, Thorpe 2003, Zedel 1991, and changed the phrasing to "*In situ studies have often…*".

– destruction?

The text here reads: "*We address specific questions about the mechanisms driving bubble presence and influence: … and the mechanism and manner of its destruction*".

We are referring to the end of a bubble: the point at which it has either completely dissolved and shrunk to nothing or has collapsed to leave no gas pocket but potentially a temporary assemblage of its previous coating.  We think that "destruction" is an adequate description, but in this case we have changed the phrasing to "termination".

– single bubble or do you mean more a group of bubbles in the statistical sense?

The text here reads "*We address specific questions about the mechanisms driving bubble presence and influence: how and when bubbles are transported downwards from the surface, how the size and number of bubbles varies with conditions, the overall path of a bubble through the water column, and the mechanism and manner of its termination*".

We are not sure what the reviewer is referring to, but we are assuming it is the phrase "*the overall path of a bubble through the water column*".   Clearly, questions exist both about what is likely to happen to an individual bubble and also the collective consequences for an assemblage of bubbles. We do not see any advantage in making that distinction here; the point is that almost nothing is known about the likely path of any bubble and how that depends on its radius, the water conditions, the flows associated with its formation, its coating and the changes to the bubble due to dissolution. The point of our work is to shed some light on the likely processes, and the location of a bubble within the water column will have a strong influence on what happens to it (as our results show).  As the reviewer has pointed out, this paper is already long, and we have tried to make it as concise as possible while getting the essential points across.  We consider that the quality of the paper is best served by leaving the text as it is.

– platform. Was there a platform atop the buoy?

"Platform" is commonly used in the literature to refer to any structure that carries instruments at sea – here the spar buoy.  However, to avoid ambiguity here, we have modified this sentence to read "… and a downward-pointing foam camera mounted at the top of the buoy".

– I think there is some analysis before averaging… Also, where are the analysis algorithms described? As written Al-Lashi 2016 and 2018 are only citations for the camera.

We are not sure what the reviewer is referring to, but we think that it is the processing of the bubble camera data. The full procedure is described in the companion paper Czerski et al 2021, and Al-Lashi 2016 gives a full description of the image processing techniques used (and the image processing is the focus of that paper). There are no further processing stages, other than averaging to 1 s data (the data is recorded at 16 Hz).

155 – was the depth of the bubble camera with respect to waves known? I presume so, but it is not stated except in a statistical sense.

As stated at the start of the methods section, the bubble camera was 2m below the water level in flat water, and as stated in the following sentence, full deployment details are available in the companion paper (now accepted for this journal). The wave wires did provide a high time-resolution measurement of the water level relative to the buoy, which gave a local measurement of waves. However, we did not use this data to calculate instantaneous depth because this required very high confidence in the time synchronisation (within a fraction of a second) between the camera system, and the wave-wire data. The deployments were several days long, making such synchronisation hard to achieve. We have therefore chosen to limit the description to a statistical one based on the wave-wire data, since this is the most certain and robust measure.

170 - $10^{-9}$ is a number $10^{-4.5}$ is a weird number. Two lines later you write $2 \times 10^{-7}$, why not write $10^{-6.99}$? and elsewhere. We both know the answer.

We have used the convention for void fraction which is consistent with the most relevant figure for that part of the discussion. Here, the use of $10^{-4.5}$ is used because it is relevant to figure 3: we split the bubble size distributions in a logarithmic way, and it is more intuitive to consider intervals as: $10^{-8} – 10^{-7.5}$; $10^{-7.5} – 10^{-7}$; $10^{-7} – 10^{-6.5}$ etc than to give intervals as $1 \times 10^{-8} – 3.16 \times 10^{-8}$ ; $3.16 \times 10^{-8} - 10^{-7}$, which is also inaccurate because of the rounding necessary. However, in the following lines we are discussing data from the resonator, which has a narrow range and is best considered on a linear scale in figures 5 and 6, so we choose the format $2 \times 10^{-7}$. We consider that the current usages are the clearest and most concise way of describing the evidence needed for each set of results. As the reviewer points out, any reader who wants to convert between the two will be perfectly capable of doing so. To make the consistency clear, we have altered the phrase "Void fractions at 4 m varied from $10^{-8}$ (the noise level) to $2 \times 10^{-7}$" to "Void fractions at 4 m varied from $1 \times 10^{-8}$ (the noise level) to $2 \times 10^{-7}$"

172 - If the noise is $10^{-8}$, then how do you measure $10^{-9}$? Are there different noise levels at different depths? Why and please report nose levels for all depths.

The camera at 2m depth could measure a void fraction of $2.8 \times 10^{-10}$ in principle (one bubble at the limit of the resolution seen on only one of the frames needed to average over one second) and the resonator at 4m could record a minimum void fraction of $10^{-8}$ as stated in the methods section and in this section. In practice void fractions at 2m as low as that level were never seen, but we have added a note on the minimum detectable void fraction for the camera in the methods section. (line 157).

A comma is not appropriate here.  The sentence reads "*We did observe "plumes" (we use the term here to indicate bubbly regions several metres in size with void fractions at 2 m which were above $10^{-6}$),...*".  If the comma is included, it implies that any bubbly region with a void fraction at 2m is a plume, and that it is incidental that those void fractions are also above $10^{-6}$. That is not the case: it is a critical part of the definition that a void fraction above $10^{-6}$ is a necessary condition for this region to be considered a plume. We have made one small grammatical change (replacing "which" with "that") and the sentence now reads "*...several metres in size with void fractions at 2 m that were above $10^{-6}$*".  We have left the grammar unchanged.

– 177 – you mention 2 m depth, then 4 m depth, then its unclear if you did see plumes at 2 m and none at 4 m or you saw at both.

The void fraction distributions and the likelihood of bubbles being detected at 2m at the same time as at 4m is discussed in depth in the companion paper. It is not appropriate to repeat that discussion here, particularly when this paper is already very long.  The first sentence of this paragraph (directly before lines 172-174) reads "*Measured void fractions at a depth of 2 m ranged from $10^{-9}$ to $10^{-4.5}$, with a sharp cut off at the higher limit; detailed descriptions of void fraction results are given in Czerski et al. (Czerski et al., 2021).*".  We consider that it is already clear that a detailed description of the void fraction results are discussed elsewhere and that there is no gain by repeating further details here.

- void faction probability distribution – maybe define this? What does the word smooth mean in this sense (not a technically defined)? what time scale are the probability distributions assessed over? I realize this is in the companion paper, but please either restrict to just the needed info for this paper (why is smooth relevant), or provide more detail.

 We consider that "void fraction probability distribution" is a self-explanatory term: it is a probability distribution function describing the relative likelihood of different void fractions being observed in specific conditions.  "Smooth" has a well-known technical mathematical description: it applies to a function that is differentiable everywhere and therefore continuous. This applies to the void fraction probability distributions: there was no jump that could separate a "plume" from a "non-plume" region.  This is fully discussed in the companion paper and we do not think that it is appropriate to lengthen this paper by repeating that discussion here.  However, we have changed the wording to "*The probability distributions of the void fraction...*".

– Figure 1 please plot (a) as semiology or loglog. Here it clearly says normalized (how and why)

 We do not see any advantage in converting Figure 1(a) to a log-log or semi-log plot, and the reviewer has not suggested a reason for doing so.  As shown in figure 1(c), the maximum bubble radius ranges from 20-500 microns, which is unsuitable for plotting on a log scale. Probabilities range from 0-1 (with a maximum normalised value here of around 0.35) and it is conventional and most useful to plot them on a linear scale.  In addition, normalising data in this context involves dividing it by the area underneath the curve, something which is more complicated to calculate if the intervals are logarithmic rather than linear.  Normalised in this case refers to dividing the total number of entities in each radius bin by the total number of entities in all bins. We do it here because (as implied by the caption) there were vastly different numbers of cases in each wind range bin; this is a standard statistical procedure. We have added the word "functions" to the phrase "probability density functions" at the start of this paragraph to clarify the definition.

figure says normalized. How was it normalized? Why was it normalized – how much does the PDF increase in reality? The normalized suggests that perhaps 0-100 μm might decrease– but I suspect not. Worst case, please put un-normalized data in supplemental. How is the PDFor histogram calculated – uniform bins?

We have used "normalised" to imply the standard probability density function (which is a normalised distribution) and there was no additional normalisation.  We have changed the caption to read "Probability density function" for clarity.

Why is there less data at high wind speeds? Maybe a PDF of wind speeds could be useful. In any case, if you aggregate 22 and 26 m s-1 bins (Why such weird wind speed bins) youhave 122 photos but a clearly decreased maximum size at 100% probability. So this explanationis possible, but not that convincing, especially given how the camera is quite shallow a nontrivial amount of time.

There is less data at high wind speeds because the wind blew at those speeds for less time over the expedition.  We could only record data in the conditions seen during our time at sea. We have included all the data from the whole expedition to maximise the usefulness of the analysis, and it would be inappropriate to exclude data to keep the number of data points constant. The wind speed time series is shown in the companion paper, and it is typical for an expedition of this sort: there is considerable variation over time as storms blow through.

The wind speed bins are all separated by 4 m/s in integer multiples of 4 (so 12-16, 16-20, 20-24 etc) and they are plotted at the centre of each bin (14, 18, 22).  This is a perfectly conventional procedure. We show the 100% data for completeness, but we give higher weight to the lower percentiles of the distribution because they are less likely to be skewed by single anomalies (in this case, 2 photographs out of 1.5 million). Our explanation is consistent with the broad pattern shown by the lower percentiles.

– 201 - Seems more of a discussion section paragraph than a results paragraph.

This paragraph does include some discussion, but the reasons we set out in our general response at the top of this document.  There are many sets of results to present, and if we leave all discussion until the end, the reader will have to go back and forth to figures to check on relatively minor points.  It is also the case that the nature of these results motivate the analysis done in section 3.2, and we want to give that motivation here. We consider that some preliminary analysis is helpful at this point, and the discussion at the end can focus on the larger synthesis.

205 – its not really an artefact – which is where an analyzer fails, it is the result of time averaged size distributions being affected by a bubble-size dependent residence time. Please rephrase.

An artefact is defined by the Oxford English Dictionary as "*something observed in a scientific investigation or experiment that is not naturally present but occurs as a result of the preparative or investigative procedure*". The feature we describe here is not naturally present but occurs as a result of the processing that is commonly done. We consider the word "artefact" to be an excellent description of this situation and we have not made any changes.

215 - It would be clearer to just state what the new ranges are. I also see that whereas the unnormalized ranges are similar across radii, the normalized ranges are not as consistent. Additionally, 4m, station 3 normalized clearly is two part and should be modeled accordingly. If you define a variable in the caption for the slopes, why are you not using it in the text?

We consider that the critical point is the width of the range, and since this is shown on a log scale, the quickest and simplest way to do that is by giving the factor over which the distributions vary. The point here is not the absolute numbers but their variability.

We are not sure how to interpret the comment about normalised and unnormalized ranges being similar across radii: this is the same data, and all points in each individual distribution are normalised by the same number. We have presented the data and shown our analysis of it, but no data set falls on to a perfect pattern.

We do not agree that 4m station 3 is "clearly two-part". It has a complex shape, and we show the fitted slopes to help with the interpretation of the plot, not as a statement that we think all the data falls along that line. The line is also not a model: it is a consequence of data analysis. In the case of the 2m data, there is a much clearer pattern. There are also subtleties in the interpretation of the resonator data for the smallest bubble sizes (which are discussed in other papers but not here), and this means that limited weight should be given to those smallest bubble sizes here. The data is useful, but differences in the bubble coating can cause significant differences in interpretation of the details. We have therefore focussed the fit on the most robust part of the distribution, which is the bubbles larger than 20 microns in radius, and note that this is consistent with the 4-m data at the other stations.

We assume that the comment about the variable in the caption refers to S1 and S2, which are also labelled on some figure parts – these are merely labels for parts of the figure, not 'variables' in a mathematical sense. We think that it is important to show these individual slopes, but they do not merit specific discussion in the text. They are intended to help show readers how individual fits varied between deployments, to provide evidence which readers may wonder about or find useful. But we have chosen to analyse the data predominantly in terms of the environmental conditions, rather than separating it by deployments. The data is all openly available, as shown in the data availability statement, and interested parties may wish to conduct their own analysis.

The sentence reads "*The normalised size distributions at 2 m have a broadly consistent shape, which can be fitted as two straight lines with a slope break at approximately R = 80 μm*".   The phrase "a slope break at approximately R = 80 μm" is a single clause, and separating the final part would make the sentence harder to understand and would add ambiguity.  We have left the text as it is.

218-219 this is repeating line 215. Delete, or rewrite. Define low bubble number (note, I don't see the lack of collapse in the data)

Line 215 reads:  "*This collapses the data, reducing the range by approximately a factor of 5 at 2-m and a factor of 8 at 4-m.*"

Line 218-219 reads: "*The void fraction normalisation collapses the bubble size distributions to a much narrower range in all cases except those with very low bubble numbers.* "

There are two key points which the second sentence adds. The first is a judgement (distributions collapsed to a much narrower range) which follows on from the numbers given in the first sentence. The second point is the final phrase: "except those with very low bubble numbers".  We think that it is important to point this out – it is notable that 2m station 3 has very low bubble numbers and does not collapse to the same pattern as the others. In addition, the lowest distributions in the plot for 2m, station 7 also do not fit the pattern.  We cannot give a specific number range because it is dependent on radius, but we think it is evidence that there are some distributions that fall a long way below the rest (never above).

We have added text to clarify where the low bubble numbers are, so that it now reads: *"... except those with very low bubble numbers (for example at 2m, station 3).*"

221 – true that dispersion of bubbles is important, but very small bubble change size by dilution very rapidly, so I don't see how you can rule that out. At higher densities, they dissolve slower (due to saturation) and at lower densities, small bubbles dissolve faster. Agreed that largerbubble size change is not dominant. Please revise.

The line here reads "*This implies that the bubble size distribution is relatively stable, and that variations in void fraction are dominated by this stable population diffusing outward in space rather than individual bubbles changing size.*"

We do not understand the phrase "very small bubble change size by dilution very rapidly".  Bubbles can lose gas by dissolution, and this may be what the reviewer meant.  We note that our data is not consistent with the idea that bubbles of different sizes are growing or shrinking at different rates, and that is the point of this section of the results.  If smaller bubbles were growing or shrinking at different rates to larger ones, the bubble size distributions could not be so consistent in shape at all times.  The implication of the results we show here (the collapse of bubble size distributions when normalised by void fraction) is specifically that the bubbles cannot be growing or shrinking in a size-dependent way.

We also note that there are no studies of the way that bubble shrink or grow in situ in the ocean – all experimental studies have been done with relatively simple coatings in laboratories. As has been shown by the work of Wurl and others, the natural load of surfactants and organics in the ocean is highly complicated.  We do not therefore think that it is appropriate to rely in our analysis on assumptions about what bubbles do in the ocean based on laboratory studies of limited relevance.  Our aim is to interpret our data without prior assumptions, and in this case the implication of the near-stationary bubble size distributions is that bubbles are not growing or shrinking significantly.  This is the conclusion that we state, justified by the evidence we show in this section.  We have left the text here as it is.

– Fig. (not Fig/)

This has been corrected.

noise or variability?

 The sentence here reads: *"..and because there is more noise in the data for small radii"*.  In the case of the resonator at these low void fractions, "noise" is the correct term – the system generates a low level of electrical noise due to its structure and electronics (which is discussed in previous work on this device, for example Farmer 1998). The signal-to-noise ratio of the system is more significant at high frequencies (which corresponds to the measurement of smaller bubbles) because the absolute level of the signal for small bubbles is lower.

252-263 – this reads more like discussion, where other arguments can be made. Suggest reducing to one sentence or just deleting here, particularly sentences that say "critical question"

This paragraph gives a very brief overview of the results from this section and uses it as motivation for the next set of results.  If this paragraph is removed, the reader is left without motivation for the analysis that follows.  We end this paragraph with the text: *"A more detailed analysis of how the gas volume is distributed across bubbles of different radii can address that question, because a fitted peak volume is a more sensitive measure of small changes in bubble size."*.  The fitted peak volume is the analysis that follows.  We consider that this short paragraph is necessary and so we have left it in place.  As we have said above, our aim with this long paper is to make sure that it's as readable and concise as possible, and we consider that signposting phrases like this improve the logical flow of the paper.

delete "positions" – you already said radii

This has been corrected.

– It seems to me (by definition) due to the bend to a slope greater than -3 (S2=-3.5), this bend occurs in the uncollapsed data as well. Suggest deleting this sentence.

The point of this sentence is not to compare collapsed and uncollapsed data.  The point is that the fitted volume peak data is consistent with the bubble size distribution results shown earlier.  We have shortened the sentence so that it now reads "*This is consistent with the normalised bubble size distributions described above.*".

279-280 – agree about the replacement, not certain about the conclusion. What about uniform dissolution and higher void fraction being from more intense wave breaking not more evolved plumes? And in any case, it seems that timescales are important for this conclusion and not mentioned for this (should be in the discussion) sentence. BTW my first through is the radius of the kink has more to do with turbulence velocities and the rise velocity of a 60-80 μm bubble, with rise velocity increasing rapidly leading to low sensitivity in r with respect to turbulence. Just a thought, and in any case, the place to discuss these issues is in the discussion. Note, the smaller possible radius of the bend-over at 50 μm as in 4m station 3, is consistent with decreased turbulence motions further from the interface.

In the companion paper, we show evidence which shows that individual waves breaking at the surface are locally decoupled from the deeper plumes - they are not linked in time or space except at the largest scales (i.e. in high wind conditions there are more breaking waves overall and therefore more bubbles). The reviewer seems to be suggesting that deep long-lasting plumes are being fed directly by waves breaking on top of them, and we are not aware of any evidence that supports this view.

It is not clear what "uniform dissolution" refers to – does it suggest that bubbles lose gas in a way that is only proportional to their radius, surface area or volume?  All of these would show up as changes in the bubble size distribution which are not seen (as the previous section describes).  We have already shown (earlier in this paper in Figure 1) that turbulent processes are not the dominant limit on bubble sizes – if that was the case then the maximum bubble size would vary with wind speed. We discuss bubble rise times later in this paper (while covering the slight uptick in the largest bubble sizes in figure 10), and we conclude that there is no evidence that bubbles as small as 60-80 microns are rising through the rest of the bubble population in a significant way.

– bubble destruction? Not sure what this means as written. Do you mean loss? Dissolution? Size change is certainly not destruction.

We are referring to the end point of a bubble: the point at which is ceases to be a bubble. We are not talking about size changes except in the case that they may precede bubble termination. We have changed the wording throughout the paper to "termination".

295 – what does wider range of conditions mean? I guess from the rest of the para that you mean higher and lower winds? Please report what the winds are for all three data sets in the text.

The higher and lower winds are referred to in the sentence that follows this first sentence, and the specific conditions shown are detailed in the figure caption. We consider that it would be misleading to be more specific than "conditions" because although the wind speeds shown are different (and probably account for the major differences), other conditions (water temperature, swell conditions, prior history) are also different for the different deployments. Full details of the wind speed history for each deployment are given in the companion paper, and we do not consider that it is helpful to repeat that information here.

301-309 –move to discussion (could varying turbulence levels explain?)

As we have discussed above, varying turbulence levels cannot explain the patterns we see here. We have left this section here to be consistent with the structure outlined above: each results section includes a brief discussion where this is useful to lead into the analysis that follows it.

Fig. 7 weird ylabel (why not use a more standard label as in Fig. 6.

We have changed the y-label on figures 7 and 8 so that they are consistent with each other, and they now read "Radius of volume peak ($\mu$m)".

and 322 peak radius as on 318. Please be consistent.

We have changed line 321 and 322 to read "volume peak radius".

– Suggest "far greater variability below"

We have made this change.

– 336 – discussion. Again, could it be weaker turbulence velocities for weaker wave breaking allowing larger bubbles to escape and only keeping smaller bubbles in suspension? Move fist sentence into next paragraph.

No, for reasons that we have already given.  The bubbles that we see at 2m depth do not immediately follow wave breaking – they are decoupled by processes in the top metre or so of the ocean. Bubbles are moved to a depth of 2m and below by coherent circulations such as Langmuir cells, not turbulence.  This is consistent with the results of Zedel & Farmer, which clearly show the spatial distribution of deep plumes linked to large coherent circulation patterns.

We are not sure why the reviewer is suggesting moving the last sentence of this paragraph. Everything in this paragraph relates to Figure 7(a), as does this final sentence. The next paragraph concerns Figure 7(b), and that final sentence is not directly relevant to that material.

 The sentence reads "*At this depth, it is clear that individual patches of bubbles each follow their own pattern, since the progression for each individual plume can be seen.*".

We think that the original phrasing was very clear, but we have changed it to "*is visible*".

351-352 – please explain why it is clear that individual patches (two dimensional – don't you mean blobs?) each follow their own pattern? Do you mean trajectory? As written it makes no sense. Do you mean each bubble plume has a distinct pattern?

We have not used the word "blob" anywhere in this paper, as we consider it colloquial.  We have changed the wording to "*At this depth, it is clear that each individual plume of bubbles has a distinct relationship between void fraction and peak volume radius, clearly clustering along discrete curves. The progression for each individual plume is visible: each group of markers along one diagonal line represents a single plume, and the relationship varies between plumes.* ".

 We have changed the wording to "*We note that…*".

359 – since significant non-linearities might be expected near the surface – how was the extrapolation done? What curve fit

Oxygen percentage saturation data is almost always displayed after conversion to the percentage saturation that the water packet would have at the surface (rather than at the measurement depth). This is the extrapolation we refer to – it is a standard procedure and depends on pressure, salinity and temperature, with only a pressure adjustment needed in this case. Extrapolation is necessary here because the CTD did not start measuring until it reached a couple of metres beneath the surface (the measurement is at timed intervals, so the exact depth depends on when the timer is switched on).  In practice, most papers would show Figure 9(b) as oxygen saturation without further comment, ignoring the implicit extrapolation.  A CTD unit will carry out this calculation automatically.  We have changed the phrasing to *"... and the saturation state at the surface, inferred from the highest available measurement in the water column"*.

We argue at the end of this paper that more attention needs to be given to local spatial variations in gas saturation, but we did not have these measurements available here.

– how do you know the surface ocean was always undersaturated? Doesn't this depend on your extrapolation and its accuracy? And surely you don't mean even in surfacing bubble plumes (as you note in the next paragraph).

As we have stated above, the extrapolation is a standard conversion so that saturation percentages are given relative to surface saturations.  All the CTD measurements showed oxygen concentrations that were below the saturation concentration at the surface and therefore definitely below the saturation concentration at any other depth.  Consequently, the surface ocean in general was always undersaturated, as expected at this time of year.  As we note, this data is limited because it lacks spatial detail, but we think that it is valuable to show what a standard CTD cast measures in this situation.

– extrapolated is not observed.

As explained above, in this case it is.  The statement is accurate as given.

– surely you could find a citation on this matter of saturation from waves. I will see if you have one in the discussion….

We could not find any previous reference to this idea, and we think that it is very important to raise it here.

Section 2.5 – limitations – this is a discussion topic.

This section comes directly before the Discussion starts. We think that it is valid to describe the limitations of results before discussing their possible meaning.  It is not possible to come to conclusions about data without being aware of its limitations, and these limitations apply to the whole data set and so are best set out in one place before the discussion starts. We have left the text as it is.

– and found no bubbles (missing verb)

We have made this correction.

"appear to be too low for the limiting factor to be the balance between buoyancy and turbulent flow."

We are not sure what point the reviewer is making here. Deane's previous paper suggested that the maximum bubble radius would be set by the balance between buoyancy and turbulent flows. Our results, presented in Figure 1, show that the maximum bubble radii we found are far lower than that balance would suggest. This is what this sentence reiterates.

– accumulation of surfactants also likely plays a role. In any case, surface bubbles are advected downwards anywhere as vertical fluid motions are by definition zero at the sea-air interface (except for breaking waves).

We are not aware of any literature that suggests that bubbles accumulate significantly more in regions of high surfactant load. This is speculation that we are not willing to add to this paper, particularly since we lack direct surfactant measurements. We disagree that surface bubbles must be advected downwards – they are mostly likely to remain at the air-sea interface, advected only horizontally in the frame of reference of the surface.

it is not the bubble size distribution that is advected – it is the bubble plume with a specific bubble size distribution that is advected. Suggest laterally rather than sidewise which is colloquial.

We have changed the phrasing to *"…and the bubbles are advected sideways"*.

– destroyed? Maybe dissolved.

As we have stated, there is no direct evidence from the ocean on the speeds of bubble dissolution or the mechanism that finally ends a bubble. We are avoiding making assumptions which are not based on previous in situ ocean studies, and so we do not think that "dissolved" is the right word to use here. We have changed the wording to "terminated".

– pulled downwards? Suggest advected downwards

The sentence reads "*and contain long-lasting bubbles that were pulled downwards by previous advection patterns, and possibly also bubbles mixed downwards gradually by turbulence*".

We do not want to use the word "advection" twice in a single clause. We have changed it to "moved".

– delete gradually – you don't know if it is or is not gradual, and in any case, what is gradual? Not defined.

The Oxford English Dictionary defines "gradually" as "*By a gradual process; little by little; by degrees*". It does not necessarily imply a timescale, but does imply a continuous process rather than a discrete one. The conclusion of this work is that the fastest process moving bubbles downward is advection by the downward limbs of Langmuir circulation. The organised patterns of bubble plumes seen both by us and also Zedel & Farmer cannot be due purely to bubbles advected by turbulence, since this would not cause organised patterns. Any downward advection by turbulence must therefore happen considerably more slowly than the Langmuir advection. We have left the word "gradually" in, because we think that this accurately reflects the relative timescales of the two processes. We cannot be specific about timescales because this is not known, although the companion paper does provide some evidence on downward flow speeds.

– "cannot last long enough" specify, please. Dissolve and/or rise too rapidly to form a background population.

As we have already stated, we think that use of the word "dissolved" pre-empts a full understanding of the process of bubble termination: the gas must eventually shift to a state of being dissolved in the water, but use of the word "dissolve" implies a smooth process. As we state in this paper, we think that serious consideration must be given to the idea that bubble termination is a rapid process after a long period of pseudo-stability. We also show evidence that the majority of bubbles in these plumes are not rising significantly due to buoyancy, so it is not appropriate to refer to that process – that is not what is happening at 4m depth in a plume. We do not know enough about the processes at play to say what is happening to the bubbles at 4m depth, although the evidence suggests that they disappear via a process that is not size-dependent (as we state in this section). So we agree that the wording could be considered vague, but we think that it is the most accurate and concise way of reflecting the current knowledge.

– what do you mean by heterogeneity of the bubbles? Normal meaning is spatial heterogeneity; however, this makes no sense in this sentence.

We are indeed referring to spatial heterogeneity. There is no reason for a convergence zone to be completely filled with bubbles at a uniform void fraction: that would be a strange outcome considering that the wave-breaking process is stochastic in nature. If the bubble void fraction is not uniform in space, and if it is patchy in a way that reflects bubble production processes, it must be spatially heterogeneous. We are describing a situation where bubbles are produced at random positions and times at the surface, each generating a small contribution to the surface bubble layer, and that this spatially heterogeneous layer is then advected horizontally until it reaches a convergence zone. At that point, it will be advected downwards to form a plume, and the heterogeneity of the bubbles in the plume will reflect the heterogeneity of bubbles in the surface layer. We do not think that this is a controversial idea. We have left the text unchanged.

487-494 this feels like a repetition of text earlier in this section. In fact, I think this whole section can be re-organized and shortened and better focused.

This section is the end of the Conclusion, and we are summarising some of the most important results from this work. We think that this point bears repeating in this context, particularly for readers who jump straight to the Conclusion. We have not made any changes.

– note, when bubbles burst at the sea surface it creates new microbubbles.

One study (Bird, 2010) observed microbubbles formed in laboratory conditions when perfectly symmetrical surface bubbles burst. This study was carried out in a viscous fluid (water–glycerol–sodium dodecyl sulphate solution), and its relevance to ocean studies is not clear. We are aware of no studies that show this happening in the ocean, or claiming that it is a significant process.

– can you provide some guidance on what is a large bubble (r>??)

The phrase "large bubbles" when discussing breaking waves is commonly taken to imply bubbles with a radius greater than approximately 1mm. We have added text to clarify this here.

– you are not really suggesting anything. Perhaps better to explain this as a knowledge gap – proposing an unknown process or processes or series of processes but not proposing what they are is not actually proposing something.

 The text reads as follows: "*We suggest that there is an unknown series of processes in the top metre or so of the ocean which convert the highly unstable initial population with void fractions ~$10^{-1}$ into a pseudo-stable size distribution which can persist for at least several minutes, and which has a maximum void fraction limit of $10^{-4.5}$.*".

We are indeed identifying a knowledge gap, while constraining it with the evidence that we have gathered in this study. The consistent maximum void fraction implies that a consistent set of processes is responsible for the characteristics of the bubble populations reaching a depth of 2m. Our data suggest that these processes are limited to the top metre only, and this is an important statement to make. The bubble populations we see at 2m cannot be explained by the current (very limited) models for what happens to bubbles once formed by a breaking wave, and so we are suggesting that this set of near-surface processes needs to be considered specifically. It is extremely challenging to make robust measurements of bubbles within the top metre of stormy seas, and we think that it is important to prioritise these measurements in the future. We have left the text as it is.

– mix – I think you mean diffuse….

We think that "mix" accurately represents our intended meaning in this sentence, but for clarity we have changed the wording so that it now reads "*This may occur over many minutes as bubbles partially dissolve, are lost from the population as they rise under buoyancy, or may be mixed and advected by both turbulent and coherent motions while remaining close to the ocean surface.*".

516-518 – citation needed (e.g., the work of Bruce Johnson)

We have added references to Johnson & Wangersky and Poulichet & Garbin. and also to Chua, Chitre & Deane in the following sentence.

– probably also the type and concentration of surfactant – not just its presence or absence – particularly given how much surface area is created by a bubble plume.

We have amended "surfactant presence" to "the presence and composition of surfactants and particulates".

– destroyed is what a military does to a target with a missile! And if you check out the Johnson paper, you do not know if the bubble really is destroyed or shrinks to a stabilized microbubble of a few microns in a surfactant matrix. Just say dissolve or burst at the sea surface.

We consider the process of termination to be important, and as stated above, we do not think that "dissolve" is the right term to use, because the current usage of that term in the literature refers only to slow dissolution. We have changed "destroyed" to "terminated" here.

525-527 – these two sentences are repetitive and could be combined and shortened.

We do not consider these sentences to be repetitive. They make two separate points: i) there is a lack of evidence for upper limits on bubble lifetime and ii) our data show that the lower limits on lifetime could be long: of the order of several minutes. We have made no changes.

– why would there be a thin bubble free layer below the sea surface? This seems unlikely given that buoyancy does not disappear. Since it is speculative with no data and seemingly nonphysical,
suggest delete.

We make no suggestion of a "bubble free layer" (perhaps the reviewer has misread this sentence). We have made no changes.

We have changed the wording to "advect".

537 – only bubbles with rise velocities less than the downwards velocity.

The question of bubble rise velocity is complex (see Ashwanth et al 2020, Mazzitelli 2003, Ruth 2021 and many others), both because of bubble shape deformation and the effect of having a collection of bubbles in close proximity. We therefore do not want to make an over-simplistic statement here. We have changed the wording to "*sufficiently small* bubbles in that water mass will be carried with it".

548 – what is the evidence that you larger bubbles are not trapped in the centre than edges. If this was in the companion paper it needs to be cited. Also how much larger is larger? This is a very qualitative assessment.

The text here reads "*The higher downward velocity in the centre of the downward limb of a Langmuir cell could trap larger bubbles than the edges, but we see limited evidence for this*."

We think that the existing phrasing makes our point clearly: faster downward velocities are expected along the centre line of the downward leg of Langmuir flow, and if that is the case, they **could** trap larger bubbles than the regions slightly to the side. This is a relative statement – no quantification of "large" is possible without reference to the velocity distributions in a particular Langmuir cell. But we make the point clearly that we do not see any evidence for this here. We have made no changes to the text.

No. We have discussed this point above.

The phrasing we have used is standard English. There is no grammatical issue with the phrase "*could also be*", and in this case it is the clearest and most concise construction. We have made no changes.

556 – what about stabilized microbubbles? More accurate to say the vast majority or almost all, or something not so definitive. Also write in present tense, not passive past tense. Microbubbles could diffuse to the sea surface from those depths. Plus Langmuir circulation is

The text here reads "*It seems clear that once bubbles are advected to 2 m (and the depth limit could be even shallower), the gas they contain will eventually all be dissolved into the ocean. None of our evidence supports the idea that bubbles might rise from these deep plumes to return to the ocean surface.*"

As we state clearly, our results do not support the idea that bubbles may return to the surface once they have descended below 2m depth. The dramatic difference in void fraction (a factor of 100) between depths of 2m and 4m demonstrates that bubbles are destroyed more rapidly at depth (and the large change over only two metres depth cha**n**ge supports the idea that this is a rapid process, not a slow one). Langmuir cells are thought to be approximately square in cross-section (Thorpe, 2004) and Farmer & Li 1995 saw cell horizontal separations of 10-20 metres in wind speeds of 10-15 m/s. Even in these relatively mild condition (by the standards of our own study), water packets were being carried to depths of 5-10 metres before they return to the surface. In the case of our study, it seems clear that water packets from the surface are being carried far deeper than 4m, and that the vast majority of bubbles will have terminated long before a water packet is carried sideways and then upwards again. There is no indication that any significant numbers of bubbles will make it back to the surface, even by diffusive processes (which will be slower than the Langmuir circulations). We therefore think that the definitive statement as we have given it is robust and we have made no changes to the text.

**562 – forms**

The text reads "…*the bubbles in the stabilised population which form the deeper plumes do not have a significant influence on the oxygen content of their host water mass*.". The sentence is describing the effect of the bubbles, plural, and therefore "form" is the correct grammatical construction. We have made no changes.

**562 – all these bubble are injected into the ocean? They are already injected into the ocean!**

We have changed the phrasing to "*although the gas in these bubbles will all ultimately be injected into the ocean…*".

**556-569 – very long and convoluted sentence.**

Lines 556 – 569 cover two paragraphs and 8 sentences, none longer than 2.5 lines. We consider that the language is clear here and have not made any changes.

**579 – as written this is not correct. The force on the bubbles is always in equilibrium. Furthermore, its hard to see how bubble dissolution could decrease saturation state. A very**

clumsy sentence grammatically. In any case, I am not certain if this section is really needed – it doesn't make many points and the one that it does – mixing lowers void fraction – which could be merged into other discussion sections.

The text here reads as follows: "*At increased depths the external pressure will cause the inwards force on the bubble surfaces to increase and the actual saturation state of the water around them to decrease relative to its initial saturation state.*".

There are two clauses here and both are correct. Both refer to the effects of pressure, in two separate ways. As depth increases, the static pressure in the liquid surrounding the bubble will increase because of the increased weight of water above. This force is exerted on the external wall of the bubble. The bubble may shrink in response to that pressure or withstand it without changing size if there the coating of particulates and surfactants has become rigid. In both cases there could be, but is not necessarily, some exchange of gas across the bubble boundary. The second clause refers to the difference between the saturation state at the surface and the saturation state at depth *for the same water packet*. As pressure increases, the gas concentration in the liquid required for 100% saturation increases, so the saturation state of the water can change as it moves in the water column, without any change in actual gas concentration. We think that the phrasing here is clear, and expresses the intended point concisely.

-or they are produced rapidly and dispersed widely or transit to deeper depths and thus have limited residence time at this depth – Farmer ad Li (1995) show bubbles from Langmuir circulations to 10 m.

The text here reads "*There are many regions at 4 m depth even during very high winds when the bubble void fraction is well below $10^{-8}$, which suggests that high production rates and relatively short lifetimes are more likely*". The reviewer seems to be making the same point that we have done in this sentence: that the evidence supports the idea that bubbles are produced at a relatively high rate, and have a limited lifetime as soon as they start to move downwards. We note that Farmer's observations were carried out with a very sensitive high frequency sonar which would respond very strongly to extremely small numbers of resonant bubbles (which would necessarily be very small: of the order of 10 microns or below). There is thus no contradiction between their results and ours: extremely low void fractions would be visible on the Farmer sonar data, far lower than the void fractions we measure here, but would be insignificant for gas transfer. We have made no changes to the text.

– we suggest that bubbles collapse . . . This is speculative, so please propose a new mechanism as dissolution is quite well understood and is not consistent with a sudden collapse. Or delete. Additionally, any talk of sudden and slow needs to specify what timescales are slow or sudden.

As we have stated, we are aware of no in situ measurements in the ocean that show bubbles dissolving slowly until they disappear, although most models assume that this is the case. We have noted that several experimental studies (Chua 2021 and others cited above) discuss bubbles which are stabilised and not changing size.  We are presenting our results without reference to prior assumptions, and we think that an important point to come from our work is that these long-lasting ocean bubbles are not changing size slowly.  We have made no changes to the text.

600-605 – The increase of hydrostatic pressure with depth (and decreasing saturation) is not new and is not a process that would be termed sudden collapse.

This section of the paper is setting out our understanding of the whole train of events, providing the necessary context without claiming that all of it is novel.  It would weaken the paper to omit the mention of this physical process here.  In addition, we have not stated that the increase of hydrostatic pressure is the same as sudden collapse – one may cause the other in a non-linear way, but that is not the point that we are making here.  We have left the text as it is, because it conveys the necessary scientific points in a concise way with sufficient context.

- Open questions should not be after conclusion! This seems more like a future research section in the discussion. Also, all these points have been brought up throughout the discussion section and even the paper, so this is a bit of a repetition – I think it is good to summarize them all somewhere, but then lessen the number of times these are mentioned throughout the paper.

Our aim is to make it easy for future researchers to identify future priorities.  It is entirely appropriate for this list of open questions to follow the conclusions: we have set out the scientific judgements that we have made based on our own results, and here at the end we summarize the questions which emerge as priorities following our work.  They are referred to earlier in the paper, but separately.  We think it is valuable to bring them together at the end.  We have made no changes.

i. This is a long open question – maybe break into two. You have measurements of two environmental conditions – introduce with your findings and then be more specific on your open question

This is an umbrella question: one over-arching priority, followed by specific examples from the work presented in this paper.  We consider this to be the clearest way of making this point and have not made any changes.

ii. Couldn't you answer this with sonar data? At least roughly?

No. The limitations of single frequency sonar data are well-known, some of which are set out (when relevant) in this paper. This technique cannot provide bubble size distributions or void fraction measurements, and it also provides no data on bubble movement in the water column.  So there is no way to assess the overall population near the water surface and how it is moving from our sonar data, particularly for bubbles which are within a metre of the rapidly-changing wavy surface.

iii. timescales of advection at convergence zones. – you report 20-40 s, so this is not an open question or it is poorly phrased.

The timescales of advection have not been measured in any detail.  We inferred that 20-40 seconds could be the time it takes for bubbles to travel from 2m to 4m based on our ADV data, but that provides no information about the rest of the Langmuir cell and is a very limited measurement.  More detailed measurements are needed to make strong links between gas fluxes and plume presence.

vi. nature of bubble coating – this was really not addressed in the paper, so seems beyond your paper's scope.

It is indeed beyond the scope of this paper, but it is a significant open question in this field.  Our results add to the weight of evidence suggesting that more knowledge about the nature of bubble coatings and their effects on bubble dissolution/collapse processes is essential.  At the moment, this is the source of much speculation, but there are very few in situ observations, and we wish to highlight it as a priority.

Viii (where did vii go?)

We have corrected the list number.